# Transient viral exposure drives functionally-coordinated humoral immune responses in HIV-1 post-treatment controllers

Luis M. Molinos-Albert[1,2], Valérie Lorin [1,2], Valérie Monceaux [3], Sylvie Orr [4], Asma Essat[4], Jérémy Dufloo [5], Olivier Schwartz [5], Christine Rouzioux[6], Laurence Meyer[4], Laurent Hocqueloux [7], Asier Sáez-Cirión [3], Hugo Mouquet [1,2✉] & ANRS VISCONTI Study Group*

HIV-1 post-treatment controllers are rare individuals controlling HIV-1 infection for years after antiretroviral therapy interruption. Identification of immune correlates of control in post-treatment controllers could aid in designing effective HIV-1 vaccine and remission strategies. Here, we perform comprehensive immunoprofiling of the humoral response to HIV-1 in long-term post-treatment controllers. Global multivariate analyses combining clinico-virological and humoral immune data reveal distinct profiles in post-treatment controllers experiencing transient viremic episodes off therapy compared to those stably aviremic. Virally-exposed post-treatment controllers display stronger HIV-1 humoral responses, and develop more frequently Env-specific memory B cells and cross-neutralizing antibodies. Both are linked to short viremic exposures, which are also accompanied by an increase in blood atypical memory B cells and activated subsets of circulating follicular helper T cells. Still, most humoral immune variables only correlate with Th2-like circulating follicular helper T cells. Thus, post-treatment controllers form a heterogeneous group with two distinct viral behaviours and associated immune signatures. Post-treatment controllers stably aviremic present "silent" humoral profiles, while those virally-exposed develop functionally robust HIV-specific B-cell and antibody responses, which may participate in controlling infection.

[1] Laboratory of Humoral Immunology, Department of Immunology, Institut Pasteur, Université Paris Cité, Paris 75015, France. [2] INSERM U1222, Paris 75015, France. [3] HIV, Inflammation and Persistence Unit, Department of Virology, Institut Pasteur, Université Paris Cité, Paris 75015, France. [4] Centre de Recherche en Epidémiologie et Santé des Populations (CESP), Université Paris-Sud, Université Paris-Saclay, INSERM, Le Kremlin-Bicêtre, France. [5] Virus & Immunity Unit, Department of Virology, Institut Pasteur, Université Paris Cité and CNRS UMR3569, Paris 75015, France. [6] Assistance Publique-Hôpitaux de Paris, Service de Microbiologie Clinique, Hôpital Necker-Enfants Malades, Paris, France. [7] Service des Maladies Infectieuses et Tropicales, CHR d'Orléans-La Source, Orléans 45067, France. *A list of authors and their affiliations appears at the end of the paper. ✉email: hugo.mouquet@pasteur.fr

Certain HIV-1-infected individuals, called post-treatment controllers (PTC), maintain controlled viremia after interrupting antiretroviral therapy (ART), often when initiated shortly after primary infection[1]. By stopping early HIV-1 replication and spread, ART limits viral reservoir seeding and deleterious infection-induced inflammation, preserves innate immunity, and protects T-cell and B-cell compartments from irreversible damage[2–7]. Such beneficial effects likely participate in the HIV-1 post-treatment control[8]. Yet, the immune mechanisms contributing to post-treatment control are not fully elucidated[8,9]. PTC are reminiscent of natural controllers who spontaneously achieve long-term control of infection in the absence of ART[10]. However, as opposed to spontaneous controllers, PTC neither frequently carry the protective HLA class I alleles (i.e., HLA-B*57 and HLA-B*27), nor possess highly efficient cytotoxic CD8+ T lymphocytes[10–12]. Coordinated and functional antibody B-cell responses often develop in natural HIV-1 controllers[13]. In fact, elite controllers who are characterized by suppressed viremia conserve high frequencies of circulating follicular helper T (cTfh) cells, and HIV-1-specific memory B cells[14,15], and produce polyfunctional antibodies equipped with cross-neutralizing and Fc-dependent effector activities[16–21]. Sustained viral antigenic stimulation and diversification in very low viremia settings such as in natural controllers can thus support the development of broadly neutralizing antibodies (bNAbs) targeting the HIV-1 surface envelope glycoprotein (Env)[22]. Hence, a potential role of humoral immunity in the slow progression and natural control of HIV-1 infection has been proposed[13]. Early ART also preserves blood and gut HIV-1-specific B cells[5,23,24], but precludes the formation of potent or cross-neutralizing antibody responses upon treatment-induced viral suppression[25]. Nonetheless, early ART may facilitate a rapid and efficient induction of neutralizing antibodies after treatment interruption (TI)[26–28]. There is scarce data in few PTC pointing towards an absence of heterologous seroneutralization[29–31]. Still, the humoral response to HIV-1 in infected individuals who received early therapy and control infection after ART discontinuation has not been investigated in detail.

Here, we use an integrated immunoprofiling approach combining measures of serological, B-cell and cTfh-cell parameters with clinico-virological variables to dissect the humoral response to HIV-1 in the largest described cohort of long-term PTC. Our results reveal a heterogeneity in PTC who segregate into two distinct immuno-virological subgroups. Stably aviremic PTC are characterized by a low-magnitude antibody response akin to early-treated individuals on ART. In contrast, PTC experiencing transient viremic episodes develop a multicomponent humoral immunity against HIV-1 associating the preservation of functional cTfh-cell sub-populations, the expansion of blood Env-specific memory B cells and the production of IgG antibodies with cross-neutralizing and Fc-dependent effector activities.

## Results

**Study population.** To study the humoral immune profiles in HIV-1 post-treatment controllers, 22 PTC from the ANRS VIS-CONTI cohort[1] (median duration of virological remission time post-TI: 12.6 [2.8–18] years), and 21 matched early-treated HIV-1-infected individuals from the ANRS CO6 PRIMO cohort[32], who experienced post-TI viral rebound (referred to as post-treatment non controllers, PTNC), were included in the study (Supplementary Table 1). Of note, donors 005002 and 038001 were the only PTC included in the study who did not start ART during the acute phase of the infection. PTNC were matched to PTC according to ART initiation (51.5 [26–100] days post-infection) and treatment duration (3.8 [0.48–11.17] years), but

PTNC rebounded on average within the year post-TI (Supplementary Table 1). Although some PTC remained stably aviremic during the long-term virological remission (sPTC, $n = 10$), a subgroup of PTC experienced at least one viral load >50 copies/ml during the follow-up period and were referred to as virally-exposed PTC (ePTC, $n = 12$). During the period before sampling, 20.6% [3.5–100%] of all measured viral loads (28.5 on average [3–40]) were over 50 RNA copies/ml, with low (50–400 copies/ml) to higher (>400 copies/ml) viremia being detected in 10% [0–66.7%] and 7.5% [0–53.8%] of cases, respectively (Fig. 1a, b and Supplementary Table 1). The last transient viremia >50 copies/ml was registered on average 2115 [0–5 230] days prior to sampling (Supplementary Table 1). For most ePTC, viremia consistently ranged from 50 to 400 copies/ml (056002, 063003, and 084001) or transiently above 400 copies/ml (005001, 070001, and 073001). Of note, few ePTC had frequent detectable viremia (070001, 180003, 200001, and 216001), and three of them (070001, 200001, and 216001) experienced more than two consecutive viral loads (VL) over 400 copies/ml and resumed ART by the time or before the analysis. Prior to treatment resumption, donors 070001, 200001, and 216001 were in remission post-TI for 2.1, 8.6 and 12.7 years, and showed detectable viremia in 53.8%, 46.2% and 73.7% of the analyzed samples post-TI, respectively (Supplementary Table 1).

**Comprehensive serological antibody profiling of PTC and PTNC.** To investigate the antibody response to HIV-1 in PTC sub-groups, as well as in PTNC on ART and post-TI, we analyzed up to 43 quantitative and qualitative serological immune parameters. First, we measured the ELISA binding of purified serum IgG antibodies to HIV-1 p24 and various Env proteins [BG505 SOSIP.664 trimers, subtype A (UG37), B (YU2), D (UG21) gp140 foldon-type trimers, subtype B (YU2) and C (CN54) gp120 proteins, and MN gp41 subunit]. Strikingly, ePTC showed higher IgG antibody reactivities against all HIV-1 antigens tested as compared to sPTC (Fig. 1c and Supplementary Fig. 1a). One ePTC corresponding to a particular case of perinatally-infected individual[33] (073001, Supplementary Table 1), and for whom the last detectable VL was recorded 9 years before this analysis, had the lowest antibody levels (Fig. 1c and Supplementary Fig. 1a). As anticipated[34], IgG1 was the dominating sub-class among anti-p24 and anti-gp140 IgG antibodies in PTC, and was found at a higher level in ePTC (Supplementary Fig. 1b). In contrast, serum IgG antibodies from sPTC and ePTC bound similarly to human mucin isoforms 2 and 16 (Supplementary Fig. 1c), the latter being known to interact more strongly with agalactosylated IgG antibodies from chronically HIV-1-infected individuals[35]. Strong reactivity of purified IgA antibodies to HIV-1 p24 and BG505 SOSIP proteins were only found in two ePTC (003800 and 180003) (Supplementary Fig. 1d). Next, we evaluated the impact of early ART and viral rebound on the HIV-1 antibody response by measuring anti-Env and anti-p24 IgG antibody titers in PTNC on ART and post-TI. As expected, the viral rebound post-TI in PTNC (Fig. 1d) was accompanied by an increase of HIV-1 antibody levels and neutralizing activity against the easy-to-neutralize BaL.26 virus (Fig. 1c and Fig. 1e). Of note, anti-Env IgG levels in PTNC on ART were similar or superior to those in sPTC, whereas p24 antibody titers were higher in sPTC than in PTNC on ART (Fig. 1c and Supplementary Fig. 1a). Conversely, IgG antibody titers in PTNC post-TI and ePTC were comparable for all HIV-1 antigens, except for BG505 SOSIP and p24 IgG titers which were lower in ePTC (Fig. 1c). A longitudinal analysis of the antibody responses, including pre- and post-TI timepoints, was only possible for four ePTC (063003, 084001, 200001, and 216001) and one sPTC (098004) (Fig. 1f). sPTC 098004 had globally stable anti-Env antibody titers overtime, while anti-p24

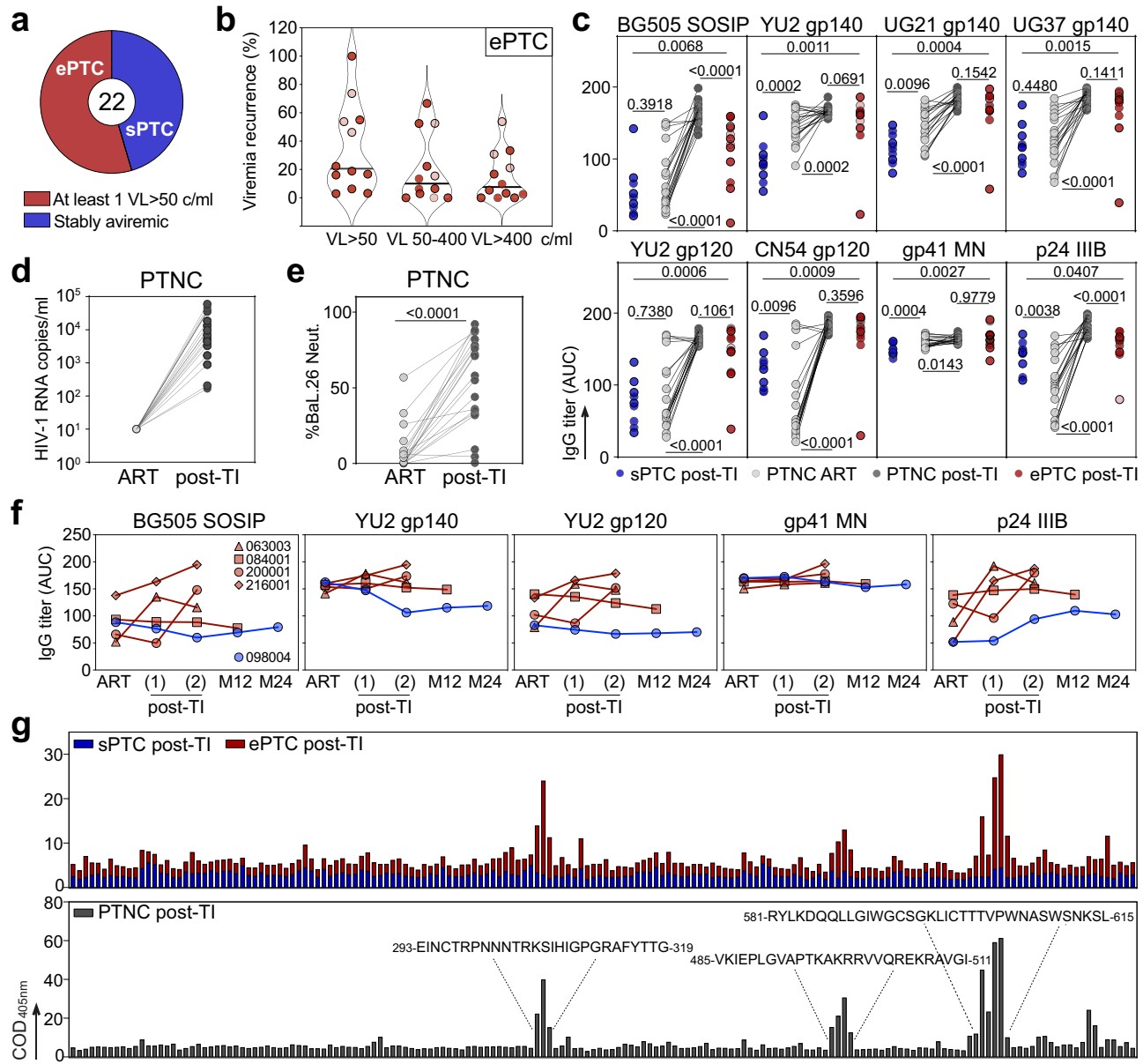

**Fig. 1 Serum HIV-1 antibody responses in PTC and PTNC. a** Pie chart distinguishes the PTC experiencing transient viral loads (VL) over 50 copies/ml (ePTC, red, $n = 12$) from those stably aviremic (sPTC, blue, $n = 10$). **b** Violin plots showing the frequency of transient viremic episodes in ePTC ($n = 12$) according to VL. Light red dots indicate ePTC who resumed ART before the analysis (Supplementary Table 1). **c** Dot plots comparing the serum anti-Env and anti-p24 IgG titers (as area under the curve of ELISA optical density values (AUC)) between sPTC (blue, $n = 10$) and ePTC (red, $n = 12$) after treatment interruption (TI), and in PTNC ($n = 20$) before (light grey) and after TI (dark grey). Groups and PTNC timepoints were compared using two-tailed Mann–Whitney and Wilcoxon tests, respectively. *p* values are indicated. **d** Dot plots comparing HIV-1 VL in PTNC pre- and post-TI ($n = 20$). Average time post-TI is 1.73 years (Supplementary Table 1). **e** Dot plots comparing the IgG seroneutralizing activity against Bal.26 HIV-1 strain in PTNC pre- and post-TI ($n = 20$). Timepoints matched those in (**c**), and were compared using a two-tailed Wilcoxon test. **f** Graphs comparing the serum IgG antibody titers against HIV-1 Env and p24 proteins over time for the selected PTC (ePTC, red, $n = 4$; sPTC, blue, $n = 1$). Post-TI (1) corresponds to 2.07 [1–3.22] years after TI, post-TI (2) corresponds to 14.2 [12.7–18] years after TI, and M12 and M24 timepoints correspond to 12 and 24 months after post-TI (2), respectively. Donors 216001 and 200001 were "rebounders" who resumed to ART at post-TI (2) (Supplementary Table 1). **g** Bar graphs showing the IgG seroreactivity against clade B Env overlapping peptides as measured by ELISA in PTC sub-groups and PTNC post-TI. Each bar shows the cumulative IgG reactivity for all donors against a single peptide. Amino acid sequences on the top indicate immunodominant linear Env regions. Source data are provided as a Source Data file.

IgG levels slightly increased during the post-TI period, which is reminiscent of a described association between high p24 IgG titers and slow disease progression[36,37]. HIV-1 IgG levels in ePTC 084001, who only experienced a single episode of low viremia (Supplementary Table 1), were higher but stable overtime (Fig. 1f). In contrast, ePTC 063003 and ePTC rebounders 216001

and 200001, had an elevation of HIV-1 antibody titers during the post-TI period (Fig. 1f). ELISA seromapping using consensus B Env overlapping peptides showed that serum IgGs bind principally to gp120 V3 loop and gp41 immunodominant epitopes for both PTNC and ePTC, while no specific reactivities were detected for sPTC (Fig. 1g).

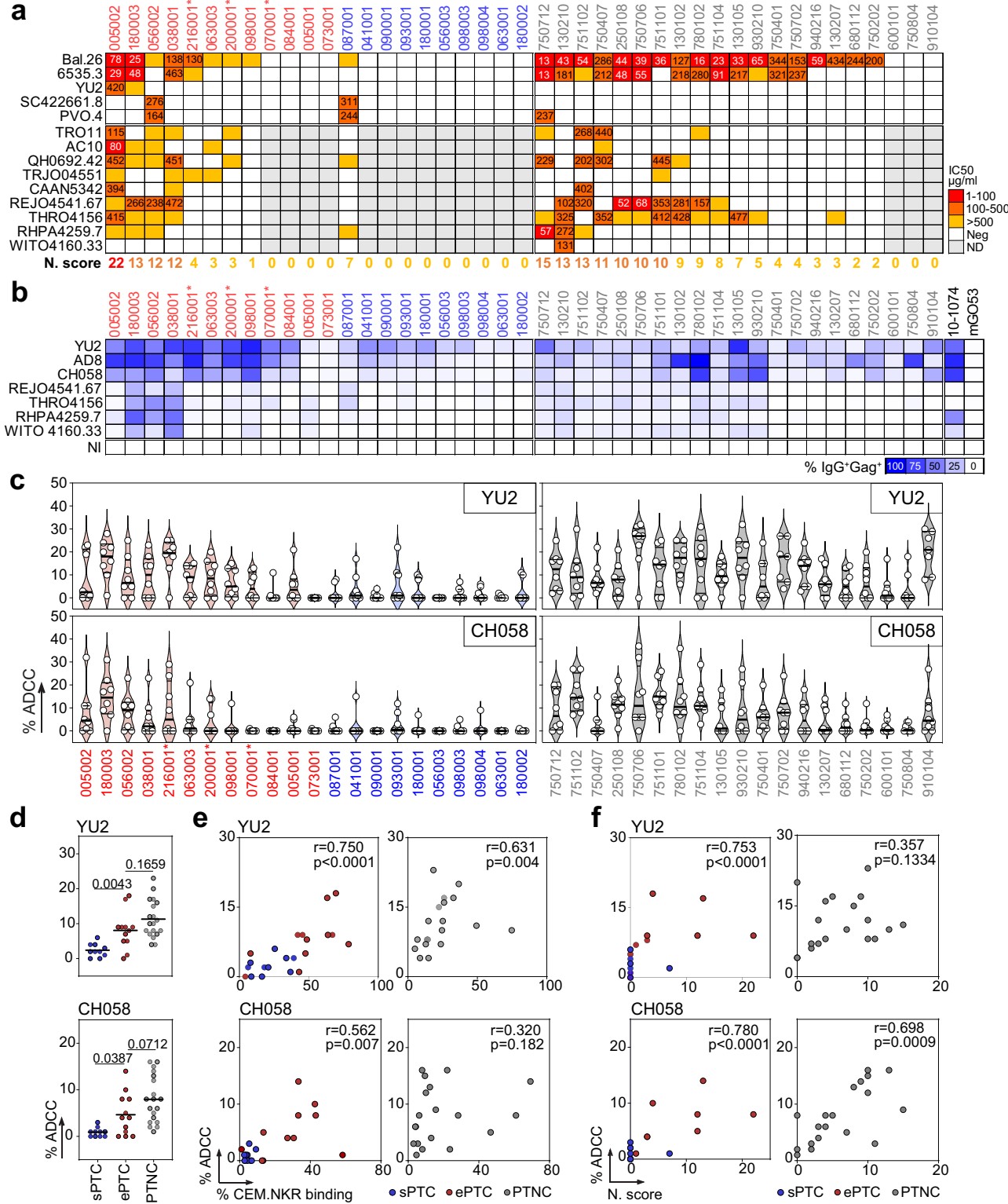

**ePTC develop robust functional antibody responses**. We then measured the in vitro neutralizing activity of purified serum IgGs using a five clade B-virus panel, and tested 9 additional tier-2 clade B strains for the samples neutralizing at least one virus in this panel. Consistent with weaker HIV-1 seroreactivities, only one sPTC (10%, *n* = 10) had antibodies neutralizing a few tier-1/-2 viruses [neutralization score (NS) = 7] (Fig. 2a). In contrast, cross-neutralization was detected in 7 out of the 12 ePTC (58.3%), including ePTC 005002 who showed a broad seroneutralization

profile (NS = 22) (Fig. 2a). Similarly, 13 of the PTNC post-TI (61.9%, *n* = 21) developed tier-1/-2 cross-seroneutralization, and 19% had NS > 10 (vs 33% for ePTC) (Fig. 2a). In contrast, serum IgA antibodies purified from PTC did not neutralize tier-1 viruses (Supplementary Fig. 1e). Serum autoreactivity has been previously associated with bNAbs development[38]. Even though purified IgGs from the PTC producing HIV-1 neutralizing antibodies did not bind HEp-2 cell-expressing antigens (Supplementary Fig. 2a, b), a significant increase of the serum IgG

**Fig. 2 Neutralizing and ADCC activity of serum IgG antibodies from PTC and PTNC. a** Heatmap comparing the IgG seroneutralizing activity against a 14-viruses panel in PTC and PTNC. Darker red colors indicate lower $IC_{50}$ values (µg/ml), while white indicates no neutralization. Grey cells correspond to non-tested purified IgG samples when not neutralizing in the screening panel (top). For each donor, the in vitro IgG neutralization score (N. score, see Methods) is indicated below the heatmap. **b** Heatmap comparing the IgG binding from PTC and PTNC to HIV-1-infected CEM.NKR-CCR5 cells. Darker blue colors indicate higher % of Gag$^+$ cells bound by purified serum IgG antibodies, while white indicates no detectable binding. The background IgG reactivity against non-infected cells (NI) control is shown at the bottom. 10-1074 and mGO53 antibodies are positive and negative control, respectively. **c** Violin plots showing the human NK cell-mediated ADCC activity of serum IgG antibodies purified from PTC [ePTC (red); sPTC (blue)] and PTNC (grey) against CEM.NKR-CCR5 cells infected with laboratory-adapted YU2 (top) and T/F CH058 (bottom) virus. Each dot represents the mean of duplicate values for a single NK cell donor. Horizontal lines indicate the mean values for all NK cell donors ($n = 8$). **d** Dot plots comparing the mean % ADCC in sPTC ($n = 10$), ePTC ($n = 12$) and PTNC ($n = 19$). Individual groups were compared using 2-tailed Mann–Whitney test. **e** Correlation plots of the mean % ADCC and % CEM.NKR cell binding in PTC [ePTC (red, $n = 12$); sPTC (blue, $n = 10$)] and PTNC (grey, $n = 19$). **f** Correlation plots of the % ADCC and neutralization scores in PTC [ePTC (red, $n = 12$); sPTC (blue, $n = 10$)] and PTNC (grey, $n = 19$). Two-sided Spearman rho correlation coefficients and corresponding p values are indicated in (**e**) and (**f**). Asterisks in (**a–c**) indicate ePTC rebounders. Source data are provided as a Source Data file.

antibody self-reactivity, measured on a panel of 10 human autoantigens, was detected for the PTC with a strong cross-neutralization (NS > 10) (Supplementary Fig. 2c–f). Of note, global IgG self-reactivity was higher in PTNC than PTC, and was neither influenced by the HIV-1 VL nor linked to the neutralizing activity (Supplementary Fig. 2e–g). Interestingly, epitope mapping experiments showed that a fraction of the IgG antibodies from ePTC 005002 and 038001 recognize the N332 N-glycosylation site (Supplementary Fig. 2h, i).

Elimination of HIV-1-infected cells by Fc-dependent effector functions requires Env-expressing cells to be bound by antibodies. Since the magnitude of "opsonization" has been previously correlated with NK-mediated killing capacity[39–41], we first measured the IgG binding to CEM.NKR-CCR5 target cells infected with two laboratory-adapted and five transmitted founder (T/F) viruses as a surrogate for antibody-dependent cellular cytotoxicity (ADCC) activity. As expected[39,40], serum IgGs from most PTC (both ePTC and sPTC) and PTNC bound to cells infected with laboratory-adapted strains (Fig. 2b and Supplementary Fig. 3). However, cells infected by T/F viruses were generally better recognized by serum IgGs from ePTC exhibiting cross-neutralizing activity (Fig. 2b and Supplementary Fig. 3a). PTNC showed more heterogeneous IgG binding-cell profiles. Those with high neutralizing scores presented low, but detectable, binding to T/F-infected cells (Fig. 2b and Supplementary Fig. 3b). Next, we evaluated the NK cell-mediated ADCC activity of IgG antibodies purified from PTC and PTNC against CEM.NKR-CCR5 cells infected with YU2 and CH058 viruses (Supplementary Fig. 4). In agreement with the infected cell-binding data, ADCC levels were significantly higher in ePTC and PTNC than in sPTC (Fig. 2c, d). Importantly, neutralizing, target cell-binding and ADCC activities were found to be strongly correlated in PTC, as well as in PTNC but only for T/F virus CH058 (Fig. 2e, f), suggesting a convergence of antiviral antibody functions.

To uncover potential associations between serological parameters, we next performed global correlation and principal-component analyses (PCA). HIV-1 antibody titers and functional activities, i.e., viral neutralization and HIV-1 infected cell-binding, strongly correlated in PTC but not in PTNC (Fig. 3a). For instance, serum IgG antibodies from PTC with the highest titers and neutralization scores bound more efficiently to cells infected with HIV-1 including T/F viruses (Fig. 3a). In contrast, high titers of serum IgG antibodies to HIV-1 antigens were detected in PTNC irrespective of their functional activity (Fig. 3a). Remarkably, PTC segregated into ePTC and sPTC clusters in the PCA of serological variables, with 64% of the variance reached when combining the two first principal components (Fig. 3b and Supplementary Fig. 5a). Total IgG and IgG1 titers to HIV-1 Env proteins (SOSIP, gp140 and gp120) contributed the most to the first principal component, followed by the neutralization score

and HIV-1-infected cell binding values (Fig. 3c). Subsequent variable reduction to 16 main parameters resulted in 78.6% of the variance obtained with the two first components for ePTC and sPTC (Supplementary Fig. 5b), and 82% when PTNC were included in the PCA (Fig. 3d–f and Supplementary Fig. 5c). Of note, the majority of ePTC clustered with the PTNC post-TI group, while most sPTC were asymmetrically distributed away from ePTC and aggregated with the PTNC on ART group (Fig. 3d–f and Supplementary Fig. 5c). Thus, our findings reveal a heterogeneity of the humoral immune response to HIV-1 in PTC, which is likely attributable to the viral dynamics: most ePTC developed a strong and functionally-coordinated antibody response, whereas sPTC were characterized by a more silent serological profile similar to the one of early-treated individuals on ART.

**Viral dynamics in PTC modulates the HIV-1 antibody response.** To investigate how HIV-1 infection impacts the antibody response in PTC, we performed multivariable analyses combining data from clinico-virological and humoral immune parameters. We found that the recurrence of transient viremic episodes was strongly associated with most serological parameters in PTC (Fig. 4a). Analyses on ePTC only still showed correlations between: (i) anti-Env serum IgG titers (both BG505 SOSIP and YU2 gp140-F), and the frequency of transient viremia <400 copies/ml; (ii) the neutralization score and the frequency of viremic episodes ranging from 50 to 400 copies/ml (Fig. 4a). Conversely, an inverse correlation was evidenced in PTC between the maximum VL before ART (pre-ART) and all anti-Env, but not anti-p24, IgG titers as well as infected cell-binding and neutralizing activities (Fig. 4a and Supplementary Table 1). Some of these associations were also observed in ePTC but not in sPTC and PTNC (Fig. 4a). Although pre-ART VL correlated with the frequency of transient viremia >50 copies/ml in PTC (Fig. 4b), they did not statistically differ in magnitude between ePTC and sPTC (Supplementary Fig. 6e). HLA class I alleles B*35 and/or B*53 are highly prevalent among PTC[1], but did not allow distinguishing PTC sub-groups and humoral immune profiles (Fig. 4c and Supplementary Fig. 6a–d). Segregating PTC according to pre-ART VL showed that the IgG binding levels to HIV-1 Env, as recombinant proteins and at the surface of infected cells, and neutralization scores were significantly higher for the individuals with pre-ART VL < $10^5$ copies/ml (Fig. 4d–f). These data suggest that both, lower pre-ART VL and short low viremic episodes post-TI, contribute to superior HIV-1 antibody responses in PTC.

**ePTC and sPTC present distinct B-cell phenotypic distributions.** HIV-1 infection triggers numerous B-cell dysregulations and abnormalities, including perturbations of subsets such as

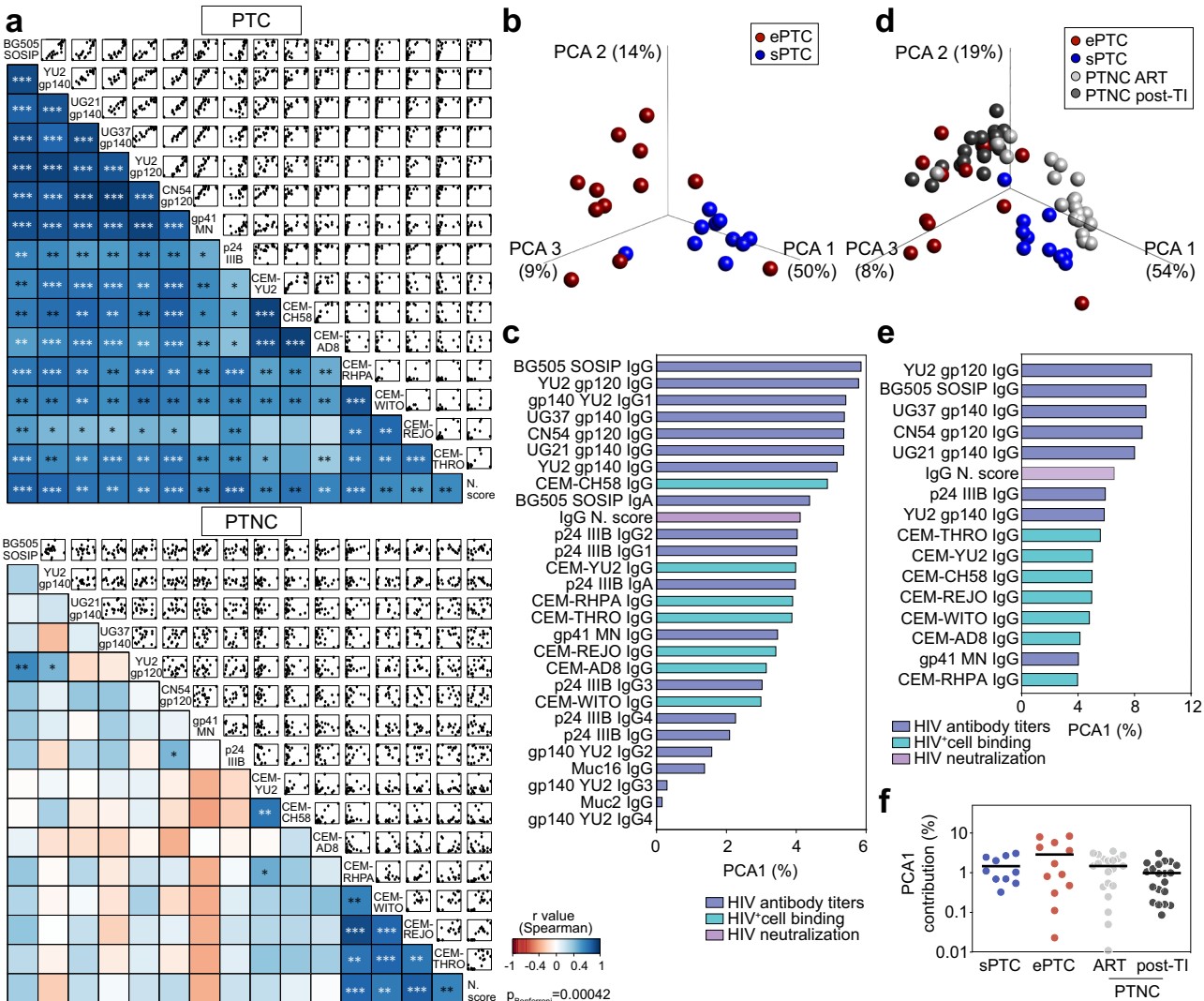

**Fig. 3 Humoral immunoprofiling of PTC and PTNC. a** Correlograms showing the correlation analyses of the humoral immune parameters measured in PTC (top) and PTNC (bottom) including IgG antibody titers, binding to HIV-1-infected cells and neutralizing activity. For each pair of compared parameters, scatter plots are shown on top and two-sided Spearman rho correlation coefficients (color coded) at the bottom. The color scale indicates the strength and direction of the correlation. Blue indicates positive correlation; white indicates no correlation and red indicates negative correlation. Asterisks correspond to unadjusted *p* values. \*\*\**p* < 0.0001, \*\**p* < 0.01, \**p* < 0.05. *p* values below the Bonferroni-corrected significance threshold (*p* = 0.00042) are highlighted in white. Detailed correlation results are presented in Supplementary Data 1. **b** 3D plot shows principal component analysis (PCA) of the humoral immune parameters measured in ePTC (red, *n* = 12) and sPTC (blue, *n* = 10) post-TI. **c** Bar graph shows the contribution to the first principal component (PCA1, as %) for each variable included in the PCA shown in (**b**). *n* = 28 variables examined over 22 samples. **d** Same as in (**b**) but including PTNC pre- (light grey, *n* = 20) and post-TI (dark grey, *n* = 20). **e** Same as in (**c**) but for the PCA shown in (**d**). *n* = 16 variables examined over 62 samples. **f** Dot plot comparing groups [ePTC (red, *n* = 12), sPTC (blue, *n* = 10), PTNC pre- (light grey, *n* = 20) and post-TI (dark grey, *n* = 20)] for the PCA1 contribution (%) of the PCA shown in (**d**). Source data are provided as a Source Data file.

expansions of transitional B cells, atypical memory B-cell populations (i.e., activated and tissue-like memory B cells) and plasma cells[42,43]. Thus, we examined circulating blood B-cell subsets in PTC by performing a comprehensive flow-cytometric immunophenotyping (Supplementary Fig. 7). First, ePTC showed a trend towards a decrease of total B cells, which was inversely correlated with the frequency of viremic episodes in the range of 50–400 copies/ml (Fig. 5a). Also, the frequency of circulating blood CD138[+] plasma cells was significantly higher in ePTC as compared to sPTC (Fig. 5b), and was positively correlated with transient viremic episodes <400 copies/ml and to the magnitude of antibody-mediated HIV-1 Env binding and neutralization in PTC (Fig. 5d). Although the frequency of transitional B cells did not differ between ePTC and sPTC (Fig. 5c), it was negatively

correlated with CD4[+] T-cell counts in PTC pre- and post-TI (Fig. 5d), suggesting that higher B-cell immaturity is driven by CD4[+] T-cell lymphopenia as shown for HIV-1 progressors[44,45].

Next, we determined the frequency of blood mature naïve and memory B-cell subsets based on the expression of CD21 and CD27 surface markers as follows: mature naïve (CD27[-]CD21[+], MN), resting memory (RM, CD27[+]CD21[+]), activated memory (AM, CD27[+]CD21[-]) and tissue-like memory (TLM, CD27[-]CD21[-]) B cells (Fig. 5e and Supplementary Fig. 7). Although RM and atypical memory B-cell populations (AM and TLM) have been shown to be respectively reduced and expanded during chronic HIV-1 infection[5,46,47], no significant differences were observed between ePTC and sPTC (Fig. 5f and Supplementary Fig. 7d). Of note, AM B-cell frequency was correlated with

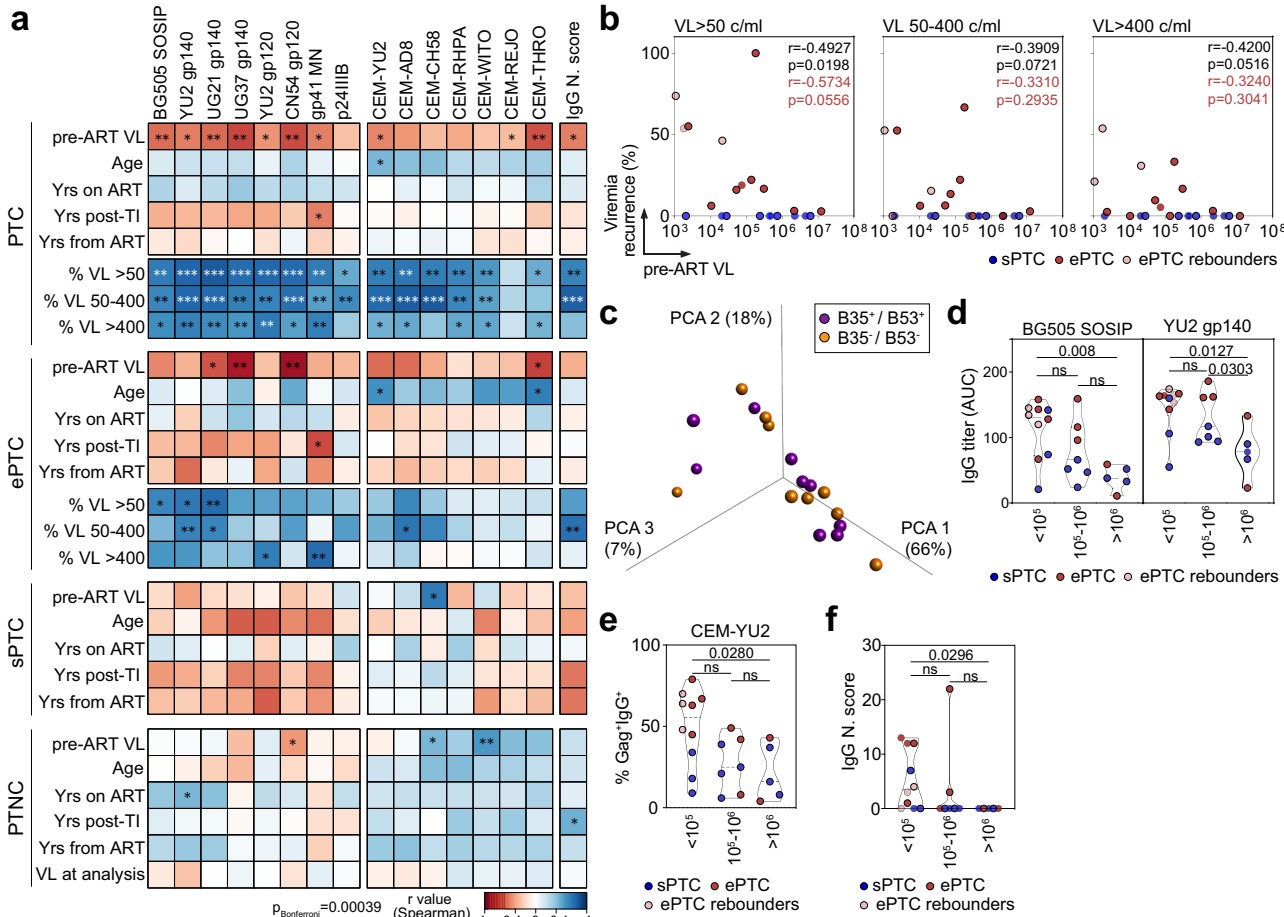

**Fig. 4 Humoral immune correlates to viral rebounds in PTC. a** Heatmaps showing the correlation analyses combining measurements for the clinico-virological [viral load before ART (VL pre-ART), age, years on ART, years post-TI, years from ART, and recurrence of viremic episodes in percent (viral load measured as RNA copies/ml (VL) > 50, 50–400 or >400)] and serum antibody [IgG antibody titers, binding to HIV-1-infected cells and neutralization score (N. score)] parameters in PTC, PTC sub-groups and PTNC. Cells are color-coded according to the value of the two-sided Spearman rho correlation coefficient (r). The color indicates the strength and direction of the correlation. Blue indicates positive correlation; white indicates no correlation and red indicates negative correlation. Asterisks correspond to unadjusted p values. ***p < 0.0001, **p < 0.01, *p < 0.05. p values below the Bonferroni-corrected significance threshold (p = 0.00039) are highlighted in white. Detailed correlation results are presented in Supplementary Data 1. **b** Correlation plots of pre-ART VL and recurrence of viremic episodes (VL > 50, VL [50–400], and VL > 400) in sPTC (blue) and ePTC (red). Two-sided Spearman rho correlation coefficients and p values are indicated for all PTC (black, n = 22) and ePTC only (red, n = 12). **c** 3D plot shows the PCA of the humoral immune parameters measured in PTC carrying HLA class I B35/B53 alleles (purple) or not (orange). **d** Violin plots comparing the IgG antibody titers against BG505 SOSIP and YU2 gp140-F trimers according to pre-ART VL values (>10$^5$, [105, 106] and >10$^6$ RNA copies/ml). **e** Same as in (**d**) but for the IgG binding of YU2-infected target cells. **f** Same as in (**e**) but for the neutralization scores of serum IgG antibodies. ePTC (red, n = 10), sPTC (blue, n = 10), ePTC rebounders (light red, n = 3). Groups in (**d**–**f**) were compared using 2-tailed Mann–Whitney test. ns not significant. Source data are provided as a Source Data file.

antibody-mediated HIV-1 Env binding and neutralization in PTC as well as separately in ePTC (Fig. 5g). Interestingly, while several ePTC displayed lower frequencies of total IgM$^+$ memory B cells as compared to sPTC (Fig. 5h), atypical memory B cells (AM and TLM) were significantly increased in ePTC (Fig. 5i and Supplementary Fig. 7d). In contrast, no significant global differences were observed when comparing the distribution of total IgG- vs IgA-switched B cells and their corresponding memory B-cell subsets between PTC sub-groups (Fig. 5j, k and Supplementary Fig. 7d). However, some ePTC showed reduced frequency of RM IgG$^+$ B cells inversely associated with expansions of AM (donors 005002 and 180003), TLM (donor 073001) or both (donor 098001) IgG$^+$ B cells (Fig. 5k and Supplementary Fig. 7d). As expected[44,45], the frequency of IgG$^+$ and IgA$^+$ RM B cells correlated positively with CD4$^+$ T-cell counts, while an increase of AM B-cell frequency was associated with lower CD4$^+$ T-cell counts and the frequency of viremic

episodes ranging from 50 to 400 copies/ml (Fig. 5l). Interestingly, the frequency of IgG$^+$ and IgA$^+$ intermediate memory (IM) B cells correlated positively with pre-ART VL, suggesting that high viremia during the acute phase effectively triggers both early germinal center reactions giving rise to IgG$^+$ IM and T-cell independent induction of IgA$^+$ IM B cells[48].

Circulating blood HIV-1 Env-reactive B cells in PTC were then quantified by flow cytometry using fluorescently-labelled YU2 gp140-F and BG505 SOSIP trimers (Fig. 5m, n and Supplementary Fig. 8a). In comparison to sPTC, ePTC displayed an increased average frequency (~3-fold minimum) of IgG$^+$ and IgA$^+$ memory B cells binding to YU2 gp140-F only or to both YU2 gp140-F and BG505 SOSIP (Fig. 5n and Supplementary Fig. 8b). Env-specific B-cell frequency correlated positively with nearly all serological parameters measured in PTC, including neutralization activity (Fig. 5o). Most Env-specific B cells had a RM phenotype, even though in some ePTC such as donor 180003, they were

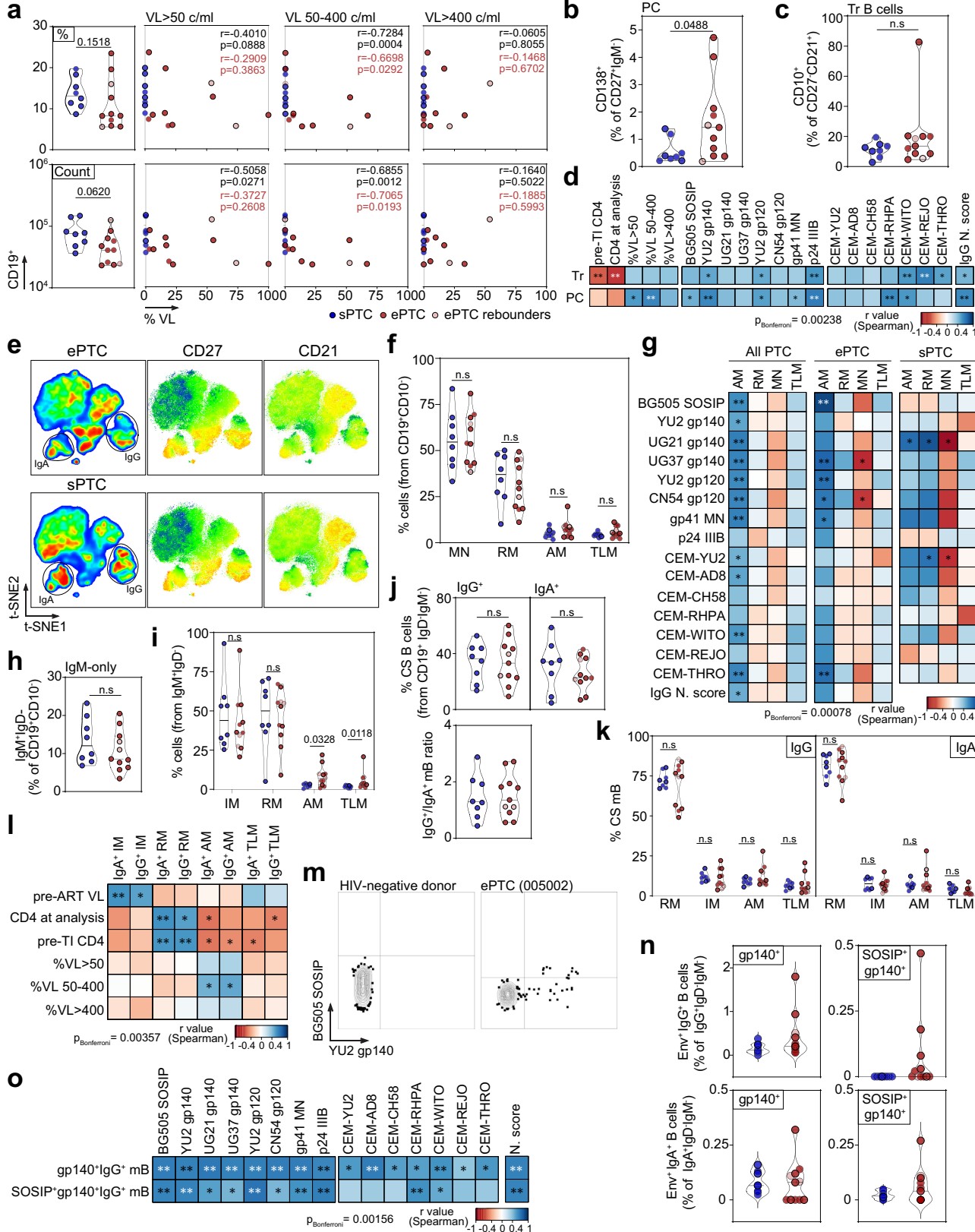

predominantly atypical memory B cells (Supplementary Fig. 8c, d). Hence, despite the fact that viral exposure markedly impacts blood B-cell compartments in ePTC leading for instance to superior cellular activation levels and expansion of atypical memory B cells, it also promotes the development of HIV-1 Env-specific memory B cells.

**Co-mobilization of activated cTfh cells and Env-specific memory B cells in ePTC.** Follicular helper T (Tfh) cells are specialized CD4+ T cells in germinal centers (GC) mandatory for T-dependent antibody responses as they support the selection of high-affinity B cell clones, and the differentiation of GC B cells into long-lived plasma cells and memory B cells[49,50]. Circulating

**Fig. 5 B-cell subsets in HIV-1 PTC. a** Violin plots comparing the % (top) and absolute number (bottom) of total CD19$^+$ lymphocytes between sPTC and ePTC. Correlation plots (right) show the % of total B cells vs viremia exposure frequency according to VL. r and p values are indicated for PTC (black) and ePTC only (red). **b** Violin plots comparing % of circulating plasma cells between sPTC (blue) and ePTC (red). **c** Same as in (**b**) but for transitional B cells. **d** Correlation matrixes of transitional B-cell and circulating plasma cell frequencies with CD4$^+$ T-cell count, viremia exposure frequency, and serological parameters. **e** t-SNE pseudocolor plots of concatenated CD19$^+$CD10$^-$ B cells in ePTC (top) and sPTC (bottom), with statistic heatmaps presenting CD27 and CD21 staining intensities. **f** Violin plot comparing CD19$^+$CD10$^-$ B-cell subset frequencies in PTC sub-groups: MN (mature naïve), RM (resting memory), AM (activated memory), TLM (tissue-like memory). **g** Heatmaps showing the correlation analyses between memory B-cell subsets and serological antibody parameters in PTC and sub-groups. **h** Violin plots comparing IgM$^+$-only B-cell frequencies between ePTC (red) and sPTC (blue). **i** Same as in (**h**) but for IgM$^+$ memory B-cell subsets. **j** Same as in (**h**) but for IgG$^+$ and IgA$^+$ memory B cells. Violin plot (bottom) comparing IgG/IgA ratios. **k** Same as in (**f**) but for IgG$^+$ and IgA$^+$ B-cell subsets. **l** Same as in (**g**) but for the clinico-virological parameters and IgA-/IgG-class switched B-cell subsets. **m** Representative flow cytograms showing blood B cells stained with Env trimers. **n** Violin plots comparing % of gp140$^+$ and gp140$^+$SOSIP$^+$ IgG$^+$/IgA$^+$ B cells between ePTC and sPTC. **o** Heatmaps showing the correlation analyses between Env$^+$ IgG$^+$ B cells and serological parameters. In violin plots, groups were compared using 2-tailed Mann–Whitney test [ePTC ($n = 11$), sPTC ($n = 8$), light red dots: ePTC rebounders]. Significant p values are indicated. ns not significant. In correlation matrixes, cells are color-coded according to r values. Unadjusted p values: ***$p < 0.0001$, **$p < 0.01$, *$p < 0.05$. p values below the Bonferroni-corrected significance threshold are highlighted in white. Correlation results are detailed in Supplementary Data 1. Source data are provided as a Source Data file.

blood follicular helper T (cTfh) cells resemble their GC counterparts[51], and are dysregulated during chronic SIV and HIV-1 infection[52,53]. To characterize cTfh cells in PTC, the frequency of CXCR5$^+$CD4$^+$ T-cell subsets as defined by the differential expression of CXCR3, CCR7, PD1 and ICOS surface markers was determined by flow cytometry (Supplementary Fig. 9a). Decreased total CXCR5$^+$CD4$^+$ T cells and increased CCR7$^+$, PD1$^+$, PD1$^{hi}$, ICOS$^+$ and ICOS$^+$PD1$^+$ cTfh-cell subpopulations were evidenced in ePTC as compared to sPTC but did not reach statistical significance (Fig. 6a, b). Frequencies of total CXCR5$^+$CD4$^+$ T cells and CD19$^+$ B cells were correlated (Fig. 6c), while the frequency of PD1$^+$CXCR5$^+$CD4$^+$ T cells was inversely correlated with the CD4$^+$ T-cell count in PTC (Fig. 6d). PD1$^{hi}$CXCR5$^+$CD4$^+$ T cells were inversely correlated with IgG$^+$ RM B cells and positively correlated with IgG$^+$ TLM B cells (Fig. 6d), suggesting a link between cTfh-cell activation and memory B-cell exhaustion. The chemokine receptor CXCR3 has been associated to distinct capacities of cTfh subsets to provide B-cell help, CXCR3$^-$CXCR5$^+$ T cells being more prone to promote class switching and affinity maturation of B cells in vitro[51,54]. In PTC, while CXCR3$^-$CXCR5$^+$CD4$^+$ T cells were predominant among cTfh cells (Fig. 6e), PD1-expressing cTfh cells (PD1$^+$, PD1$^{hi}$ and PD1$^+$ICOS$^+$) were enriched in CXCR3$^+$ cells (Fig. 6f). Despite comparable frequencies of CXCR3$^+$ and CXCR3$^-$ CXCR5$^+$CD4$^+$ T cells in both PTC sub-groups, ePTC showed a significant increased frequency of ICOS$^+$CXCR3$^+$/ CXCR3$^-$ cells as well as ICOS$^+$PD1$^+$ and PD1$^{hi}$ in CXCR3$^+$ and CXCR3$^-$ cTfh-cell subsets, respectively (Fig. 6g, h). We then performed principal component and hierarchical clustering analyses on combined B-cell and activated cTfh-cell subset data. PCA allowed segregating ePTC and sPTC, with 57.8% of the variance explained by the two first principal components (Fig. 6i). Most ePTC associated with recently activated cTfh-cell subsets (both, CXCR3$^+$ and CXCR3$^-$ CXCR5$^+$CD4$^+$ T cells), and atypical memory B cells (AM and TLM) (Fig. 6i, j), while total and IgG$^+$ RM B cells associated with sPTC (Fig. 6i). These analyses also established a correlation between frequencies of blood gp140-F $^+$SOSIP$^+$IgG$^+$ B cells and activated CXCR3$^+$ cTfh-cell subpopulations (Fig. 6k and Supplementary Fig. 9e). Thus, activated cTfh cells were mobilized in ePTC in response to brief HIV-1 antigenic stimulations, which efficiently supported the activation and expansion of Env-specific memory B-cell subsets.

**Humoral immune signatures distinguish virological profiles in PTC.** To globally unveil the humoral immune signatures linked to viral dynamics, we next performed multivariate correlation,

principal component and hierarchical clustering analyses by combining selected clinical, virological and humoral parameters including serological, memory B-cell and cTfh-cell subset measurements (Fig. 7). Frequencies of activated CXCR3$^+$ and CXCR3$^-$ cTfh-cell subsets showed common positive correlations with transient viremic episodes, transitional B-cell and plasma cell frequencies, as well as with several Env-binding B-cell and serological values (Fig. 7a). Strikingly, PD1$^+$/PD1$^{hi}$ T-cell ratio for CXCR3$^+$ but not CXCR3$^-$ cTfh cells were inversely correlated with several immunological parameters specifically, transient viremic episodes and IgG binding to soluble Env antigens and T/ F-infected cells (Fig. 7a), indicating that CXCR3$^+$ cTfh cells expressing high PD1 surface levels could be involved in the formation of stronger antibody responses. cTfh cells can be further classified according to the expression of CCR6 and CXCR3 chemokine receptors, which allow for the distinction between Th1 (CCR6$^-$CXCR3$^+$), Th2 (CCR6$^-$CXCR3$^-$) and Th17 (CCR6$^+$CXCR3$^-$) -like phenotypes[51]. The frequency of activated and quiescent Th1-, Th2- and Th17-like cTfh cells did not statistically differ between ePTC and sPTC (Supplementary Fig. 9c, d). Yet, the Th2-like cTfh-cell frequency was positively correlated with the frequency of transient viral episodes and anti-HIV-1 IgG antibody titers in ePTC (Fig. 7b), suggesting that this cTfh-cell subset was preferentially mobilized in PTC experiencing recurrent HIV-1 antigen exposures. IgG seroneutralizing activity could not be statistically associated with any of the cTfh-cell subsets, most likely due to the limited number of PTC donors displaying cross-neutralization (Supplementary Fig. 9e). Principal component and hierarchical clustering analyses confirmed that ePTC globally associated with all aforementioned serological and cellular immune parameters, whereas sPTC formed a homogenous cluster separated from these parameters (Fig. 7c, d), in agreement with their more silent immunoprofile. Collectively, our data point towards a higher mobilization of activated cTfh cells, total AM and Env-specific memory B cells in PTC individuals transiently exposed to low viral loads, who eventually developed superior polyfunctional antibody responses.

## Discussion

Investigating the virus-host interplay in HIV-1-infected individuals achieving post-treatment remission is key in identifying immune correlates of viral control and therefore, in developing novel preventive and therapeutic strategies to combat HIV-1 infection[1]. In this study, we performed a comprehensive humoral immune profiling of PTC from the well-characterized VISCONTI cohort. Our data revealed that PTC define a heterogenous group

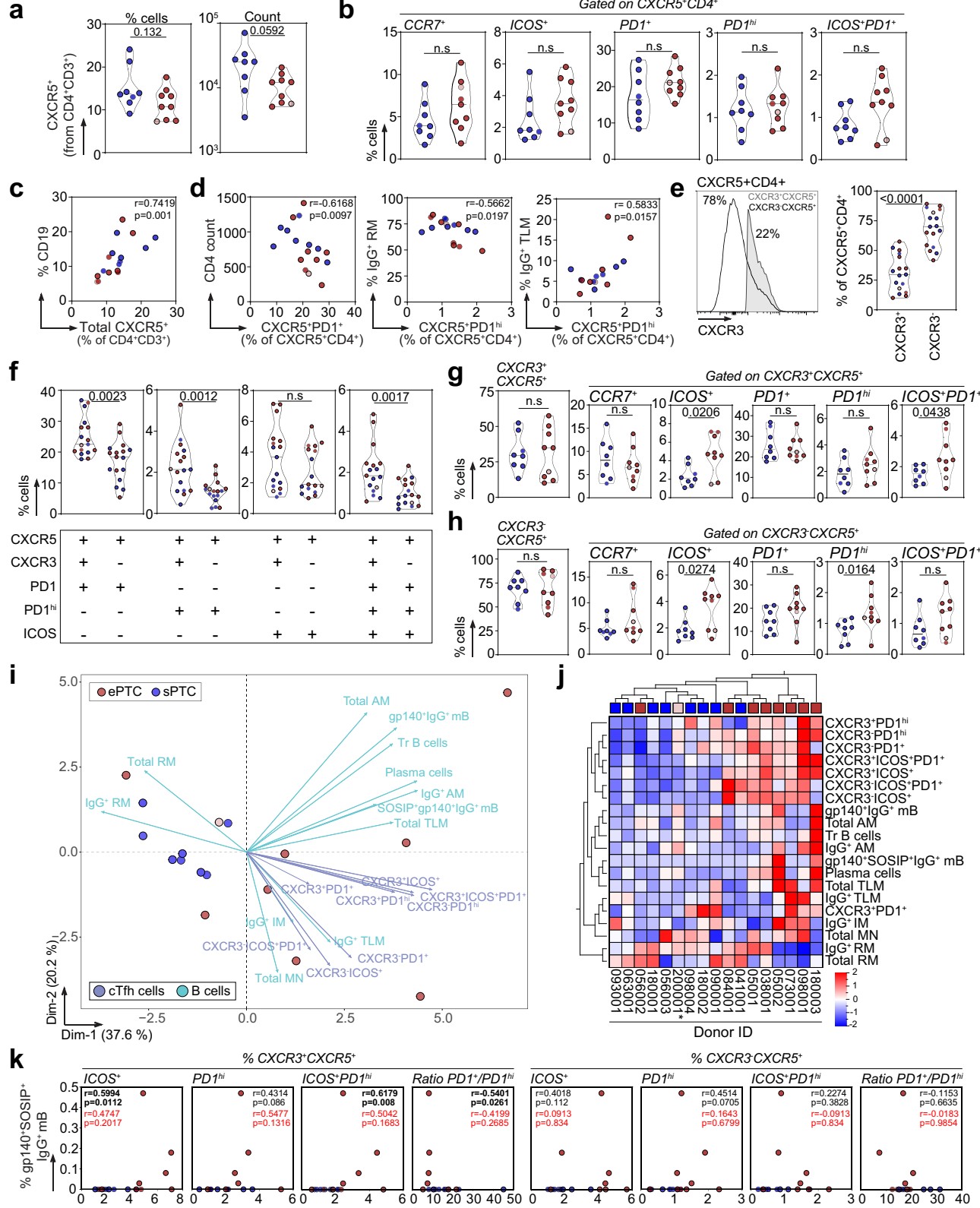

of individuals displaying divergent humoral profiles according to whether they experienced long-lasting and complete viral suppression (sPTC) or transient low-viremic episodes (ePTC) after treatment interruption. Remarkably, viral exposures in ePTC were linked to a high-magnitude, polyclonal and functional humoral response to HIV-1, as similarly observed here and elsewhere in early treated infected individuals subjected to viral

rebound post-ART interruption[27,28]. All ePTC developed high HIV-1 Gag and Env antibody levels but more importantly, one third harbored cross-seroneutralization with one individual presenting an elite neutralizer-type of profile (005002). Further investigations should be carried on the bNAbs and their targeted neutralizing epitopes that developed in the donor 005002. In addition, anti-Env IgG titers and neutralizing activities were

**Fig. 6 cTfh-cell subsets in HIV-1 PTC. a** Violin plots comparing the % and absolute number of blood CXCR5$^+$CD4$^+$ T cells between ePTC (red, $n = 9$) and sPTC (blue, $n = 8$). **b** Violin plots comparing the frequencies of CCR7$^+$, ICOS$^+$, PD1$^+$, PD1$^{hi}$ and ICOS$^+$PD1$^+$ CXCR5$^+$CD4$^+$ T cells between sPTC and ePTC. **c** Correlation plot shows the % CD19$^+$ B cells vs blood CXCR5$^+$CD4$^+$ T cells. **d** Correlation plots showing the % CXCR5$^+$PD1$^{hi}$CD4$^+$ T cells *vs* CD4$^+$ T-cell absolute number, % IgG$^+$ RM and TLM B cells. Two-sided Spearman rho correlation coefficients and corresponding $p$ values are indicated in (**c–d**). **e** Violin plots comparing blood CXCR3$^+$ and CXCR3$^-$ CXCR5$^+$CD4$^+$ T-cell frequencies in PTC. A representative flow cytometric histogram is shown (left). **f** Violin plots comparing the % blood CXCR3$^+$ and CXCR3$^-$ CXCR5$^+$CD4$^+$ T cells expressing PD1, PD1$^{hi}$ or ICOS. + and − indicates the cell surface detection of the marker or not, respectively. **g** Violin plots comparing the frequencies of circulating blood CXCR3$^+$CXCR5$^+$CD4$^+$ Tfh (cTfh)-cell subsets between sPTC and ePTC. **h** Same as in (**g**) but for the CXCR3$^-$ cTfh-cell subsets. Groups in (**a–h**) were compared using 2-tailed Mann-Whitney test. Significant $p$ values are indicated. ns not significant. Light red dots indicate ePTC rebounders in violin plots (panels **a–h**). **i** PCA 2D-plot shows the B-cell and cTfh-cell related variables discriminating ePTC (red, $n = 9$) from sPTC (blue, $n = 8$). The two first dimensions account for 57.8% of the variability. The location of the variables is associated with the distribution of the donors. ePTC rebounder 200001 is shown in red light. **j** Heatmap shows the unsupervised hierarchical cluster analysis of the variables shown in (**i**) using standardized $z$ score values. Top red and blue squares correspond to ePTC and sPTC, respectively. ePTC rebounder 200001 is indicated by an asterisk. **k** Correlation plots comparing the frequency of gp140$^+$SOSIP$^+$IgG$^+$ B-cell and cTfh-cell subsets. ePTC (red, $n = 9$) from sPTC (blue, $n = 8$). Two-sided Spearman rho correlation coefficients and corresponding $p$ values are indicated for all PTC (black, $n = 17$) and ePTC only (red, $n = 9$). Source data are provided as a Source Data file.

correlated with the antibody binding breadth to HIV-1-infected cells allowing their NK cell-mediated elimination by ADCC[39–41]. This indicates that IgG antibodies produced by ePTC possess Fc-dependent effector functions that could alter the course of HIV-1 infection[55]. This is reminiscent of the functionally-coordinated antibody responses found in natural controllers[19,20]. Conversely, sPTC showed very poor HIV-1 antibody responses, reflecting a more silent immune profile, resembling those of HIV-infected individuals under ART. These findings are congruent with viral dynamics driving humoral response to HIV-1, and confirm that temporary low-antigen exposures in ePTC can support B-cell maturation and evolution towards antibody neutralization breadth as reported in natural controllers[16–18,56–59]. In most ePTC, early treatment post-infection likely favored the formation of such functional antibody responses by preserving B cell follicle integrity and GC functions[5,24,60]. Hence, despite that viral loads and disease progression are well known to be associated with the development of neutralizing antibodies including bNAbs[59,61–64], persistent high viremia may not be exclusively required for generating cross-neutralizing antibody responses. This suggests that the repeated slow-delivery of immunogens, even at low doses, could efficiently induce high levels of neutralizing antibodies by vaccination as previously shown in immunized macaques[65]. On the other hand, transient viremia has also been associated with immunological progression such as CD4$^+$ T cell decay, loss of control and subsequent ART resumption in HIV-1 controllers[66–69]. Similarly, few ePTC experiencing viral rebound subsequently resumed to ART despite being in remission for years. Strikingly, these ePTC rebounders did not develop neutralizing antibodies, which contrast with ePTC producing functionally strong humoral responses and still controlling infection despite brief viremic periods. Hence, apart from viral exposures, additional yet-to-determine factors may exert a prominent influence on the elicitation of a vigorous and functional HIV-1 antibody response in ePTC.

HIV-1 infection strongly affects B-cell function and development leading to an abnormal distribution of memory B-cell subsets marked by a RM B-cell deficiency and conversely, an expansion of atypical B-cell sub-populations[42,43]. In agreement, the frequency of momentary viremic relapses in ePTC was associated with lower total B cells and higher frequency of AM B cells. This suggests that even a limited exposure to low viremia can lead to some developmental B-cell perturbations. We also found that CD4$^+$ T cell counts pre-TI and at the time of analysis correlated positively with RM B-cell frequency and inversely with AM and TLM B-cell frequency. This is in line with early ART initiation limiting B-cell subset alterations, and achieving a more efficient reconstitution of the RM B-cell compartment at both

systemic and mucosal levels[5,24,60]. Although no significant differences of total and Env-specific memory B-cell subsets were noted between ePTC and sPTC, the frequency of anti-Env IgG$^+$ B cells correlated with most serological parameters. This reflects a certain degree of coordination between the HIV-1 Env antibody and memory B-cell compartments in ePTC, which contrast with the fragmentary data on HIV-1 controllers reporting discordant Env-specific IgG B-cell and antibody responses[70,71]. It also illustrates the impact of residual viral replication in promoting higher proportions of HIV-1-specific memory B cells as shown for elite controllers[72]. Comparably to elite controllers and as opposed to untreated chronically infected individuals[14,73], the vast majority of Env-specific B cells in PTC had a RM phenotype, albeit some ePTC were also enriched with AM and TLM B cells. Whether RM-derived antibodies in PTC are more somatically mutated and neutralizing than those expressed by atypical memory B cells as reported for viremic progressors still remains to be determined[74].

Transient viremia exposures also impacted cTfh-cell subpopulations in PTC. Consistent with their robust proliferative capacity upon antigenic stimulation[75,76], activated cTfh-cell subsets (ICOS$^+$, PD1$^{hi}$ and PD1$^+$ICOS$^+$) were mobilized in ePTC, and associated with a shift in blood memory B-cell subset distribution. This was irrespective of the CXCR3 expression on cTfh cells that is linked to the capability in providing help to B cells in vitro. Indeed, CXCR3$^-$CXCR5$^+$CD4$^+$ T cells efficiently promote B-cell class-switching and immunoglobulin production, while CXCR3$^+$ cTfh cells are less prone to activate naïve B cells[51,54,77,78]. CXCR3$^-$CCR6$^-$ Th2-like cTfh cells correlated with viral exposure, plasma cell frequency and HIV-1-specific antibody response in ePTC as did PD1$^{hi}$ cTfh cells and particularly, PD1$^{hi}$CXCR3$^-$CXCR5$^+$CD4$^+$ T cells. This is in line with circulating blood PD1$^+$CXCR3$^-$CXCR5$^+$CD4$^+$ T cells analogous to bona fide GC Tfh cells in human tonsils being associated with bNAbs development in chronically HIV-1-infected individuals[54]. Of note, the preservation of PD1$^+$CXCR5$^+$CD4$^+$ T cells early during infection has been linked with antibody responses evolving towards broad neutralization[26]. Thus, the combination of early ART with subsequent brief antigenic stimulations in ePTC likely afford efficient Tfh-cell protection, activation and recruitment to promote more optimal HIV-1 antibody responses. We also observed a preferential expansion of PD1$^{hi}$ cells among CXCR3$^+$ cTfh cells also associated with the recurrence of viremic episodes, plasma cell frequency and HIV-1-specific antibody response. Since CXCR3$^+$ cTfh cells have been implicated in recall humoral responses such as during seasonal Flu vaccination[79], they may be associated with the maintenance of secondary responses. In this regard, transient and moderate HIV-1 antigen

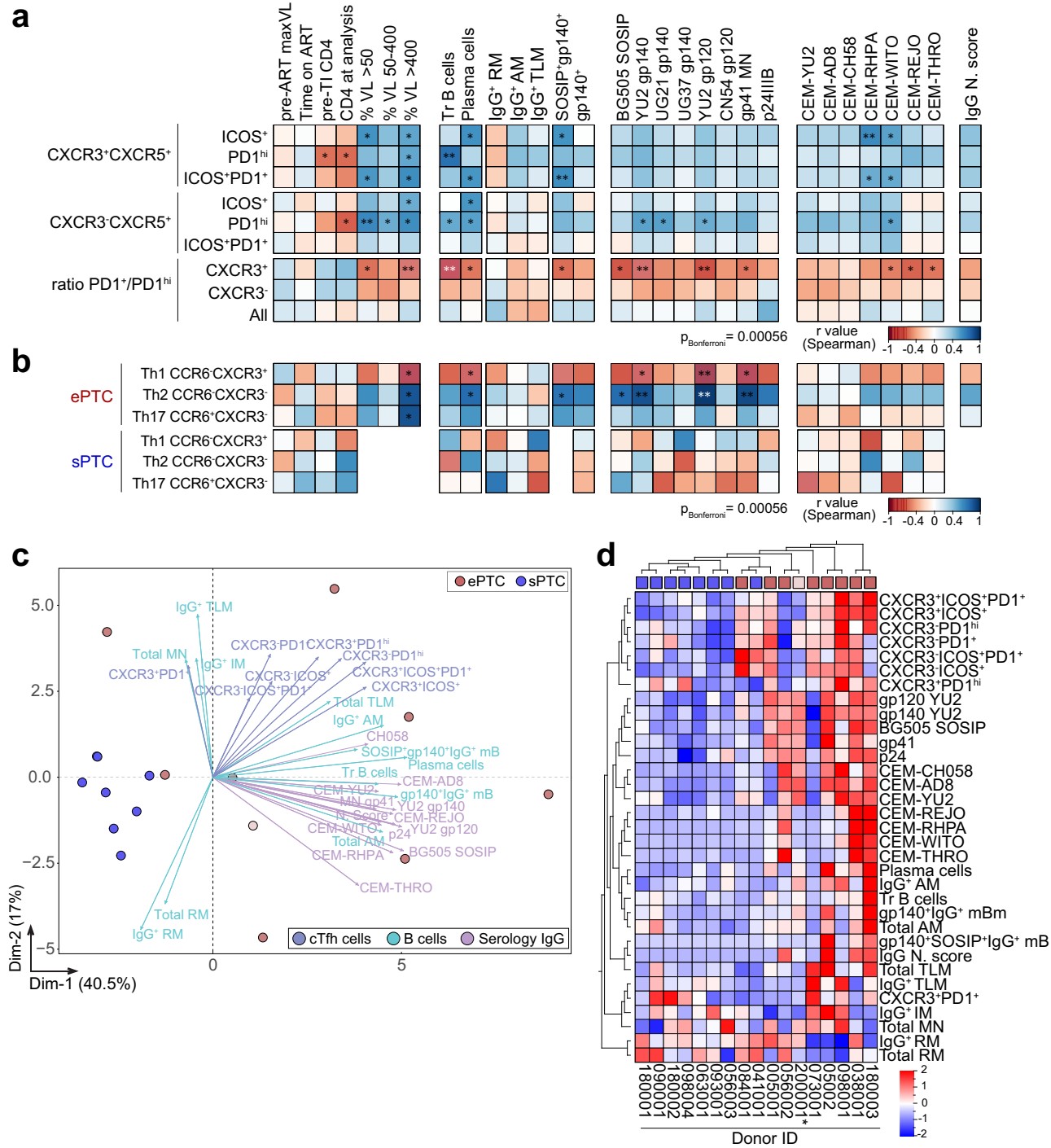

**Fig. 7 Humoral immune parameters differentiating ePTC and sPTC. a** Correlation matrixes showing the correlation analyses between activated cTfh-cell subsets (%) and selected clinico-virological and humoral parameters measured in PTC. **b** Same as in (**a**) but for the Th1, Th2 and Th17-like cTfh-cell subsets in ePTC and sPTC. Cells are color-coded according to the value of two-sided Spearman rho correlation coefficients. Asterisks correspond to unadjusted p values. ***p < 0.0001, **p < 0.01, *p < 0.05. p values below the Bonferroni-corrected significance threshold are highlighted in white. Detailed correlation results are presented in Supplementary Data 1. **c** PCA 2D-plot shows B-cell, cTfh-cell, and serological antibody related variables discriminating ePTC (red, n = 9) from sPTC (blue, n = 8). The two first dimensions account for 57.5% of the variability. The location of the variables is associated with the distribution of the donors. ePTC rebounder 200001 is shown in light red. **d** Heatmap shows the unsupervised hierarchical cluster analysis of the variables shown in (**c**) using standardized z score values. Top red and blue squares correspond to ePTC and sPTC, respectively. ePTC rebounder 200001 is indicated by an asterisk.

exposures acting as boosts could explain the expansion of PD1$^{hi}$CXCR3$^+$ cTfh cells in ePTC. Thus, distinct activated cTfh-cell subsets may be involved in the humoral response to HIV-1 in PTC, and investigating their induction, dynamics and specific

roles would require longitudinal analyses which could not be performed in this study.

The development of broad and potent seroneutralization results from the interplay of combined virological (i.e.,

"imprinting" viral strains, viral loads, viral diversification, superinfection) and host immune factors and mechanisms[80]. B-cell lineage engagement and evolution are obligatory steps in bNAbs' elicitation and maturation, and necessitate both antigenic stimulation and robust Tfh-cell help[81]. Accordingly, seroneutralization breadth developed frequently in PTC experiencing viral exposures, enabling both cTfh- and specific B-cell induction. Yet, whether neutralizing antibodies produced by certain ePTC truly contribute to sustaining viral control or whether they are elicited as a consequence of this control remains unknown. This question has been raised for natural controllers developing bNAbs and is still unresolved, even though several observations argue in favor of a role of antibodies in HIV-1 control[13]. Although the functionally-coordinated humoral response to HIV-1 in PTC appears to be a direct consequence of the viral dynamics, likely strengthened by early ART, one cannot exclude that the polyfunctional antibodies produced by ePTC have a beneficial role in controlling low viremic flares. The PTNC described here, and already reported in HIV-1 progressors with high viremia[80], can also develop cross-seroneutralization, indicating that viral dynamics can drive the humoral response independently of the HIV-1 control. Since half of the PTC experienced long-term virological remission in absence of high-quality humoral immunity, other virological and/or immune factors apart from neutralizing and/or effector antibodies are obviously involved in the infection control. This warrants further investigations to shed light on the immunovirological correlates for the long-term post-treatment control of HIV-1, and potentially reflects distinct immune pathways for achieving HIV-1 remission.

Alike natural HIV-1 controllers, HIV-1 PTC offer unique opportunities to understand the mechanisms of durable viral control off-ART, which has direct implications for designing HIV-1 remission strategies. By performing a comprehensive humoral immunoprofiling in PTC, our study reveals an unexpected divergence of the antibody B-cell response to HIV-1 between stably aviremic and low-viremia exposed individuals. Transient viral exposures in PTC trigger functionally-coordinated humoral responses featuring the development of HIV-1 Env memory B cells and potent neutralizing IgG antibodies with effector function potential, while long-term viremia suppression prevents the elicitation of efficient HIV-1 antibody responses. Altogether, our findings provide clues on the HIV-host immune interactions during long-term infection control, which upon episodic and limited viremia enable the development of potent and polyfunctional HIV-1 antibody responses.

## Methods

**Human samples**. Post-treatment Controllers (PTC) are part of the Viro-Immunological Sustained COntrol after Treatment Interruption (VISCONTI) Study of the ANRS CO21 CODEX cohort. Some of the PTC included in the present study have been previously described[1]. The inclusion criteria of PTC ($n = 22$) are as follows: (1) to present a viral load >2000 RNA copies/ml at ART initiation; (2) to be under ART during at least 12 months with suppressed viremia; and (3) to maintain a viral load below 400 RNA copies/ml for more than one year after treatment discontinuation with at least two viral load determinations during this period. The HIV-1 status and main clinical characteristics of the participants are summarized in the Supplementary Table 1. Plasma samples from 22 PTC were obtained as well as follow-up time points (month 12 and/or month 24) for certain donors. Peripheral blood mononuclear cells (PBMC) were also obtained from the blood of 17 PTC donors following Ficoll® Plaque Plus separation (GE Healthcare). Plasma samples pre- and post-ART interruption were obtained from Post-Treatment Non-Controllers (PTNC) selected from participants in the ANRS CO6 PRIMO cohort[32], who initiated ART during primary infection but experienced post-TI viral rebound. Samples were obtained in accordance with and after ethical approval from all the French legislation and regulation authorities. The clinical research protocol received approval from ethical committee (comité de protection des personnes, CPP) Tours-region Centre-Ouest 1 and Ile-de-France VII. All donors gave informed written consent to participate in this study, and data were collected under pseudo-anonymized conditions using subject coding.

**Serum IgG and IgA purification**. Human IgG and IgA antibodies were purified from donors' sera by affinity chromatography using Protein G Sepharose® 4 Fast Flow (GE Healthcare) and peptide M-coupled agarose beads (Invivogen), respectively. Purified serum antibodies were dialyzed against PBS using Slide-A-Lyzer® Cassettes (30 K MWCO, Thermo Fisher Scientific). Samples were aliquoted and stored at −80 °C.

**Antigens and antibody controls**. HIV-1 antigens: AviTagged clade B YU2 gp140[82] and BG505 SOSIP.664[83] trimers, gp120$^{HXB2core}$, gp120$^{2CCore}$, wildtype, mutant and truncated YU2 gp120 proteins[84,85] were produced by transient transfection of exponentially growing Freestyle™ HEK-293-F suspension cells (R79007, Thermo Fisher Scientific) using polyethylenimine (PEI)-precipitation method, purified by high-performance chromatography using the Ni Sepharose® Excel Resin according to manufacturer's instructions (GE Healthcare) and controlled for purity by SDS-PAGE as previously described[86,87]. The trimeric state of purified YU2 gp140 and BG505 SOSIP.664 was further confirmed by size exclusion FPLC-chromatography using an AKTA pure FPLC instrument (GE Healthcare) with a Superdex® 200 increase 10/300 GL column (GE Healthcare). For flow-cytometric B-cell staining, YU2 gp140 trimers were biotinylated using BirA biotin-protein ligase bulk reaction kit (Avidity, LLC), and BG505 SOSIP.664 directly coupled to Dylight™ 650 Dye using Dylight™ 650 NHS ester labeling kit (Thermo Fisher Scientific) according to manufacturer's instructions. Purified IIIB p24 (#12028), MN gp41 (#12027), UG21 gp140 (#12065), UG037 gp140 (#12063), CN54 gp140 (#12064) proteins, and consensus clade B 15-mer overlapping peptide library (#9480) were obtained from the NIH AIDS Reagent Program (Germantown, MD). Mucin isoforms MUC2 (MyBioSource, Inc.), and MUC16 (Bio-Techne) were obtained as purified recombinant proteins. Recombinant human HIV-1 Env-specific antibodies 10-1074[85,86], 3BNC117[84], 10E8[88], m66.6[89], and non-HIV-1 antibodies mGO53[90] and ED38[91] controls were produced by co-transfection of Freestyle™ 293-F cells (Thermo Fisher Scientific) using PEI-precipitation method as previously described[86], and purified by affinity chromatography using Protein G Sepharose® 4 Fast Flow beads following the manufacturer's instructions (GE Healthcare).

**ELISAs**. ELISA binding experiments were performed as previously described[85,92]. Briefly, high-binding 96-well ELISA plates (Costar, Corning) were coated overnight with 125 ng/well of purified proteins in PBS. After washings with 0.05% Tween 20-PBS (PBST), plates were blocked 2 h with 2% BSA, 1 mM EDTA-PBST (Blocking solution), washed, and incubated with serially diluted purified serum IgG and IgA antibodies in PBS. Binding of antibodies to overlapping linear peptides was tested using the same procedure as previously described[92]. After PBST washings, plates were revealed by incubation for 1 h with goat HRP-conjugated anti-human IgG or anti-human IgA antibodies (Immunology Jackson ImmunoReseach, 0.8 µg/ml final concentration in blocking solution), and by adding 100 µl of HRP chromogenic substrate (ABTS solution, Euromedex). For competition ELISAs, YU2 gp120 coated plates were blocked, washed and incubated 2 hours with biotinylated 10-1074 or 3BNC117 at 100 ng/ml in 1:2 serially diluted solutions of IgG competitors in PBS (ranging from 0.78 to 100 µg/ml), and HRP-conjugated streptavidin (0.8 µg/ml final in blocking solution, Immunology Jackson ImmunoReseach). Optical densities were measured at 405 nm (OD$_{405nm}$) after 1 h incubation, and background values given by incubation of PBS alone in coated wells were subtracted. All samples were tested in duplicate or triplicate, and experiments included mGO53 negative and appropriate positive controls (3BNC117 and 10-1074 for gp120, and m66.6 and 10E8 for gp41). Experiments were performed using HydroSpeed™ microplate washer and Sunrise™ microplate absorbance reader (Magellan v7.2, Tecan Männedorf, Switzerland). Self-reactivity ELISA was performed as previously described[93]. Briefly, high-binding 96-well ELISA plates were coated overnight with 250 ng/well of purified insulin (I9278), complement component C1q (C1740), Jo-1 (J4144), HSP-70 (SRP5190), histone H2b (SRP0407), proteinase-3 (SRP6309), thyroglobulin (T6830), dsDNA (91080-16-9), cardiolipin (C1649) and collagen I (C5483) (Sigma). After blocking and washing steps with 0.005% Tween 20-PBS, purified serum IgG antibodies were tested at 50 µg/ml and 7 consecutive 1:4 dilutions in PBS. Control antibodies, mGO53 (negative)[90], and ED38 (high positive)[91] were included in each experiment. ELISA plates were developed as described above using goat HRP-conjugated anti-human IgG antibodies.

**HEp-2 indirect immunofluorescence assay**. Binding of purified serum IgGs (150 µg/ml final concentration), and IgG controls to HEp-2 cell-expressing auto-antigens were assayed by indirect immunofluorescence assay (IFA) using the ANA HEp-2 AeskuSlides® kit (Aesku.Diagnostics) following the manufacturer's instructions. IFA sections were examined using the fluorescence microscope Axio Imager 2 (Zeiss), and pictures were taken at magnification x 40 with 5000 ms-acquisition using ZEN imaging software (Zen 2.0 blue version, Zeiss) at the Photonic BioImaging platform (Institut Pasteur).

**In vitro HIV-1 neutralization assay**. HIV-1 Env plasmids (Bal.26, 6535.3, SC422661.8, YU2, PVO.4, TRO11, AC10, QH0692.42, CAAN5342, TRJ04551, THRO4156, RHPA429.7, REJO4541.67, and WITO4160.33) were obtained from the NIH AIDS Reagent Program (Germantown, MD). Pseudoviruses were

prepared by co-transfection of HEK-293T cells (CRL-11268™, ATCC) with pSG3ΔEnv vector using FUGENE-6 transfection reagent (Promega) as previously described[94,95]. Pseudovirus-containing culture supernatants were harvested 48 h post-transfection, and 50% tissue culture infectious dose (TCID50) of each preparation was determined using TZM-bl cells (#8129, NIH AIDS Reagent Program) as previously described[94,95]. Neutralization of cell-free HIV-1 was measured using TZM-bl cells as previously described[94,95]. Briefly, cell-free virions were pre-incubated 45 min at 37 °C with serially-diluted purified IgG and IgA antibodies in 96-well culture plates (Costar, Corning). Mixtures were then incubated with $1 \times 10^4$ TZM-bl cells *per* well for 48 h. The 50% inhibitory concentration ($IC_{50}$) values were calculated using GraphPad Prism software (v6.0a, GraphPad Prism Inc.) by fitting duplicate values using the sigmoidal 4PL dose-response model. Samples were tested in duplicate in at least two independent experiments, which included 10-1074 and mGO53 as positive and negative control, respectively.

**Flow cytometric antibody binding assay.** Laboratory-adapted (NLAD8 and YU2), and T/F (CH058, REJO, THRO, RHPA and WITO) viruses were produced from infectious molecular clones (NIH AIDS Reagent Program) as previously described[39]. CEM.NKR-CCR5+ cells (#4376, NIH AIDS Reagent Program) were infected with inocula of selected viruses, and adjusted to achieve 10–30% of Gag+ cells at 48 h post infection[39,40]. Infected cells were incubated with purified serum IgG (50 µg/ml) in staining buffer (PBS, 0.5% BSA, 2 mM EDTA) for 30 min at 37 °C, washed and incubated with AF647-conjugated anti-human IgG antibody (1:400 dilution; #A-21445, Life technologies) for 30 min at 4 °C. Cells were then fixed with 4% paraformaldehyde and stained for intracellular Gag (anti-HIV-1 core FITC KC57; 1:500 dilution, #6604665, Beckman Coulter), as previously described[39,40]. Data were acquired using an Attune Nxt instrument (Life Technologies) and analyzed using FlowJo software (v10.6; FlowJo LLC). In each experiment, which included binding to uninfected CEM.NKR-CCR5 as a control, and 10-1074 (positive) and mGO53 (negative) controls, samples were tested in duplicate.

**ADCC assay.** Primary human NK cells were isolated from healthy donors' PBMC (Etablissement Français du Sang, France) using the MACS NK cell isolation kit (Miltenyi Biotec), and cultured in 10% FBS-RPMI supplemented with IL-2. The ADCC assay was performed as previously described[39]. Briefly, $2 \times 10^4$ CellTrace™ Far Red-stained YU2- and CH058-infected CEM.NKR-CCR5 cells were incubated 10 min at room temperature with purified serum IgG antibodies at a final concentration of 50 µg/ml. $2 \times 10^5$ purified human NK cells were added, and incubated for 4 h at 37 °C following a brief spin-down to promote cell contacts. Cells were then stained for intra-cellular Gag as previously described[39]. Positive (PGT128) and negative (mGO53) control antibodies (at a final concentration of 15 µg/ml) were included in each experiment. Data were acquired using an Attune Nxt instrument (Life Technologies) and analyzed using FlowJo software (v10.6; FlowJo LLC). The frequencies of Gag+ target cells within Far-Red+ cells were determined. ADCC was calculated using the following formula: $100 \times$ (% of Gag+ target cells plus NK with mGO53 isotype control − % of Gag+ target cells plus NK with serum IgG) / (% of Gag+ target cells plus NK with mGO53 isotype control).

**Flow cytometric immunophenotyping.** Immunophenotyping of lymphocyte populations was performed on cryopreserved PBMC samples. Cells were first stained using LIVE/DEAD fixable dead cell stain kit (405 nm excitation) (Molecular Probes, Thermo Fisher Scientific) to exclude dead cells. B and cTfh lymphocyte subsets were analyzed using two differently fluorescently labeled antibody cocktails. For B-cell phenotyping, PBMCs were first incubated with biotinylated YU2 gp140 and DyLight650-coupled BG505 SOSIP Env trimers for 30 min at 4 °C. Cells were then washed once with 1% FBS-PBS (FACS buffer), and incubated for 30 min at 4 °C with the following antibodies of the B-cell staining panel: CD19 A700 (HIB19, 1:50 dilution), CD21 BV421 (B-ly45, 1:50 dilution), CD27 PE-CF594 (M-T271, 1:100 dilution), CD10 BV650 (HI10a, 1:50 dilution), CD138 BV711 (MI15, 1:62.5 dilution), IgM BV605 (G20-127, 1:50 dilution), IgG BV786 (G18-145, 1:50 dilution), IgD APC-H7 (IA6-2, 1:100 dilution) (BD Biosciences), IgA FITC (IS11-8E10, 1:62.5 dilution) (Miltenyi Biotec), and streptavidin R-PE conjugate (1:1850 dilution) (Thermo Fisher Scientific). The cTfh-cell antibody staining panel included: CD3 BV605 (SK7, 1:50 dilution), CD4 PE-CF594 (RPA-T4, 1:50 dilution), CD185/CXCR5 AF-488 (RF8B2, 1:25 dilution), CD183/CXCR3 PE-Cy™5 (1C6/CXCR3, 1:12.5 dilution), CD196/CCR6 PE-Cy™7 (11A9, 1:50 dilution), CD197/CCR7 AF647 (3D12, 1:50 dilution) (BD Biosciences), CD279/PD1 BV421 (EH12.2H7, 1:50 dilution) (BioLegend), and CD278/ICOS PE (ISA-3, 1:50 dilution) (Thermo Fisher Scientific). Finally, cells were washed and resuspended in FACS buffer, and then fixed in 1% paraformaldehyde-PBS. Following a lymphocyte and single cell gating, dead cells were excluded. Flow cytometric analyses of stained cells were performed using a BD LSRFortessa™ instrument (BD Biosciences), and the FlowJo software (v10.6, FlowJo LLC).

**Statistical analyses.** Serological and phenotypic parameters were compared between groups using 2-tailed Mann–Whitney or Wilcoxon tests, and 1-way Kruskal–Wallis test for multiple comparisons, calculated in Graph Pad Prism

(v9.2.0 GraphPad Prism Inc). Principal component analysis (PCA) was performed using the prcomp() function in R Studio. PCA plots of individuals [fviz_pca_ind()], variables [fviz_pca_var()], and biplots [fviz_pca_biplot()], were generated using the factoextra package (v1.0.7, https://CRAN.R-project.org/package=factoextra). PCA diagrams and unsupervised hierarchical clustering heatmaps were generated using the Qlucore Omics Explorer software (v3.7, Qlucore AB). All correlograms and scatterplots were created using the corrplot and plot R functions, respectively. Correlation dot plots were generated using GraphPad Prism (v9.2.0, GraphPad Prism Inc.). Spearman rank correlation coefficients to establish multiparameter associations were calculated with a two-tailed *p* values with 95% confidence intervals in GraphPad Prism or using the R Studio Server (v1.4.1103). B-cell (gated on CD19+CD10− lymphocyte singlets) sub-populations were further visualized using the Barnes-Hut T-distributed Stochastic Neighbor Embedding (t-SNE) algorithm[96], computed in FlowJo software (v10.6, FlowJo LLC, Ashland, OR) by applying a perplexity value of 200, and 1000 iterations. Colors represent density of surface expression markers varying from low (blue) to high (red).

**Reporting summary.** Further information on research design is available in the Nature Research Reporting Summary linked to this article.

## Data availability

All statistical values for the multiparametric analyses performed and presented in the study are provided in the Supplementary data 1 file "Statistics of multiparametric analyses". Source data are provided with this paper.

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

## Acknowledgements

We are grateful to all participants who consented to be part of this study. We are grateful to Caroline Eden (New York Presbyterian-Columbia University) for helpful comments and manuscript editing. We thank Rogier W. Sanders (University of Amsterdam), and John P. Moore (Weill Medical College of Cornell University) for kindly providing BG505 SOSIP.664 expression vectors. We also thank the NIH AIDS Reagent Program (Division of AIDS, NIAID, NIH) for contributing reagents and the Agence National de Recherche sur le SIDA et les hépatites virales (ANRS) for an equipment grant support (AO2013-2 #13553, H.M.). L.M. M-A. is the recipient of a postdoctoral fellowship from the Pasteur-Roux-Cantarini program (Institut Pasteur). H.M. received core grants from the Institut Pasteur, the INSERM and the Milieu Intérieur Program (ANR-10-LABX-69-01). This work was conducted in the context of the ANRS RHIVIERA consortium, and was supported by the ANRS and the NIH/NIAID (grant #: P01AI131365, H.M.).

## Author contributions

H.M. conceived and supervised the study. L.M.M.-A. designed, performed and analyzed the experiments. V.M., S.O., A.E., C.R., L.M., L.H., A.S-C. provided human samples and associated clinico-virological data. V.L., J.D., and O.S. contributed with key reagents and expertise. The ANRS VISCONTI Study Group provided human samples and clinical data. L.M.M.-A. and H.M. wrote the manuscript with contributions from all the authors.

## Competing interests

The authors declare no competing interests.

## Additional information

## ANRS VISCONTI Study Group

Thierry Prazuck[7], Barbara De Dieuleveult[7], Firouzé Bani-Sadr[8], Maxime Hentzien[8], Jean-Luc Berger[8], Isabelle Kmiec[8], Gilles Pichancourt[9], Safa Nasri[9], Gilles Hittinger[10], Véronique Lambry[10], Anne-Cécile Beauey[10], Gilles Pialoux[11], Christia Palacios[11], Martin Siguier[11], Anne Adda[11], Jane Foucoin[11], Laurence Weiss[12], Marina Karmochkine[12], Mohamed Meghadecha[12], Magali Ptak[12], Dominique Salmon-Ceron[13], Philippe Blanche[13], Marie-Pierre Piétri[13], Jean-Michel Molina[14], Olivier Taulera[14], Caroline Lascoux-Combe[14], Diane Ponscarme[14], Jeannine Delgado Bertaut[14], Djamila Makhloufi[15], Matthieu Godinot[15], Valérie Artizzu[15], Yazdan Yazdanpanah[16], Sophie Matheron[16], Cindy Godard[16], Zélie Julia[16], Louis Bernard[17], Frédéric Bastides[17], Olivier Bourgault[17],

Christine Jacomet[18], Emilie Goncalves[18], Agnès Meybeck[19], Thomas Huleux[19], Pauline Cornavin[19], Yasmine Debab[20], David Théron[20], Patrick Miailhes[21], Laurent Cotte[21], Sophie Pailhes[21], Stanislas Ogoudjobi[21], Jean Paul Viard[22], Marie-Josée Dulucq[22], Loïc Bodard[23], Francoise Churaqui[23], Thomas Guimard[24] & Laetitia Laine[24]

[8]Service des Maladies Infectieuses, CHU Reims-Hôpital Robert Debré, Reims 51100, France. [9]Service Hématologie, CH d'Avignon Hôpital Henri Duffaut, Avignon 84902, France. [10]Service d'Infectiologie, CHI Toulon La Seyne sur Mer Hôpital Sainte Musse, Toulon 83100, France. [11]Service des Maladies Infectieuses, Hôpital Tenon, Paris 75020, France. [12]Service d'Immunologie Clinique, Hôpital Européen Georges Pompidou AP-HP, Paris 75015, France. [13]Service de Médecine Interne et Centre Références Maladies Rares, Hôpital Cochin AP-HP, Paris 75014, France. [14]Service de Maladies Infectieuses et Tropicales, Hôpital Saint Louis AP-HP, Paris 75010, France. [15]Service d'Immunologie Clinique, HCL Hôpital Edouard Herriot, Lyon 69003, France. [16]Service des Maladies Infectieuses, Hôpital Bichat Claude Bernard AP-HP, Paris 75018, France. [17]Service des Maladies Infectieuses, CHRU Bretonneau, Tours 37000, France. [18]Service des Maladies Infectieuse, CHRU Gabriel Montpied, Clermont Ferrand 63000, France. [19]Service des Maladies Infectieuses, CH Gustave Dron, Tourcoing 59200, France. [20]Service de Maladies Infectieuses et Tropicales, CHU Charles Nicolle, Rouen 76000, France. [21]Service de Maladies Infectieuses et Tropicales, HCL- Hôpital de la Croix Rousse, Lyon 69004, France. [22]Centre de Diagnostic et de Thérapeutique, Hôpital Hôtel-Dieu AP-HP, Paris 75004, France. [23]Institut Mutualiste Montsouris, Paris 75014, France. [24]Service de Médecine post-Urgence-Maladies Infectieuses, CH Départemental Vendée, La Roche-sur-Yon 85000, France.

