## [Peer Review File · Nature Communications]

Transient Viral Exposure Drives Functionally-Coordinated Humoral Immune Responses in HIV-1 Post-Treatment ControllersREVIEWER COMMENTS

Reviewer #1 (Remarks to the Author):

Overall the study includes a comprehensive assessment of multiple immune parameters associated with post-antiretroviral treatment viral control. The study focuses on two important cohorts to address critical questions related to ART-free viral suppression with implications for development of strategies to achieve and maintain viral suppression. Findings indicate immune response differences between post-treatment controllers who have transient, low level viremia [ePTC], relative to sPTC who fully suppress virus. The authors suggest that the ePCT profile in some individuals recalls the functionally-coordinated antibody responses in natural viral controllers, as well as the impact of residual viral replication in promoting higher proportions of HIV-specific memory B cells shown for elite controllers. Not surprisingly, the profile of viral control is complex, since half of the ePTC achieved long-term viral remission in the absence of high-quality humoral immunity.

- Do the authors have comments about the assays used to profile aspects of immunity, particularly if additional targets/viral protein variants might provide broader overview of the immune profile? Would research studies benefit from more sensitive viral detection methods to relate to results?
- Authors point out correctly that the “virus-host interplay among individuals who achieve post-treatment remission is key....to develop novel preventive and therapeutic strategies to combat HIV infection”. What are the implications of the authors’ findings for successful post-ART viral control by therapeutic vaccine strategy; what are implications of the results for development of a protective vaccine regimen?
- Is there a relationship between long-term ART-free viral suppression and potential HIV cure? How do the authors assess the study results for cure implications?

Specific comments.

While important scientifically, the manuscript is challenging to read and omits some essential information. Some examples [not all inclusive] are included below.

- Methods
 - Include time frame/years of the original VISCONTI study [n=22] and PRIMO study [n=21]
 - Consider harmonizing two supplemental tables into a single table with consistent presentation of data.
 - Use group designations consistently. Table S1. PTC/sPTC, post-treatment controllers [n=10]; ePTC/rebounders, in text, PTC rebounders [n=3]; ePTC, virally exposed PTC [n=9]; Table S2. PTNC, post-treatment non-controllers [n=21]. Note that the eART acronym in the manuscript uses “e” for “early”.
 - What types of ART were received? While responses are classified by CD4 and VL, the type of treatment/ART needs to be included as a variable.
 - What is definition of early treatment in VISCONTI cohort [Table S1]? While early treatment is 26 to 100 days post-infection in PRIMO cohort [Table S2], the Early Treatment designation of + or – in Table S1 is inadequate.
 - Statistical analysis methods have no set point for significant P-value. Did statistical methods include corrections for multiple comparisons? What statistical software were used?
 - Any evidence for superinfection within either of the cohorts? Were viral genotypes in PTNC or ePTC evaluated?
 - Cell-associated virus quantitation was not included in the assessment. While the assay may be unfeasible for the samples from the cohorts included in the reported study, do the authors consider such assessments of value for future studies?

- Edits

- The second section of Results extends three pages in the manuscript [p.5, line 79 thru p.8, line 158] and includes data from two figures and three supplemental figures, each with multiple data panels. The narrative would benefit from subheadings.
- Data included in Figures should be reviewed to reformat or move to supplemental. Some examples: bars in Fig 1G are essentially illegible for red/blue distribution; Fig 1 includes variant designations in different panels for the same individual [one example: Fig 1F, 84001, but 084001 in Fig 1H and Fig 1I]; blue color scale is difficult to assess in many data panels; Fig S6C: Is the method/citation for t-SNE-derived statistical heatmaps included in the Methods? Fig S1E, what is the value of the panel?
- Review English language use.

Reviewer #2 (Remarks to the Author):

This is a manuscript describing an in-depth characterization of a cohort of HIV-1 infected persons who underwent a structured treatment interruption and who were then found to control their viremia. The cohort is broken into those who had strict control and those who had blips, with the finding that those who had blips developed antibodies with breadth against various HIV-1 strains. This is a very dense paper with many multi-panel figures indicating the large amount of work that was done. Overall, the manuscript does a good job of showing that there are differences between those who have blips and those who do not.

The key take home of the paper—that people who have blips develop different responses than those who are stably aviremic—is important and contributes to our understanding of these responses, and as noted could help guide future vaccine design. I have one concern and one comment.

Concern

There is a massive amount of data in the paper, and many, many statistical tests are performed to compare groups and conditions. While the tests appear to be appropriate and properly applied, there is no discussion of correction for multiple comparisons and how this affects the analysis. I do not suggest that additional experiments be performed, but this confounder—ie, the fact that by performing so many tests a number of “statistically significant” results at $p < 0.05$ would be expected—should be discussed. Ideally, the analysis would include correction and discussion of corrected and uncorrected p-values presented.

Comment

A minor comment is made that the cohort was tested for autoantibodies by ANA and were found to be negative, contrasting these findings with those of a prior study of HIV-1-infected persons where increased autoantibody frequency was found among those who developed broad neutralizing antibodies. The paper cited did not use ANA as a measure of autoantibodies but rather used the AtheNA MultiLyte test which provides quantitation of autoantibody binding to a range of autoantigens. In full disclosure, I was a coauthor on the cited study, and we specifically did not use the ANA test because of the challenges of interpreting that test in this setting. The ANA test is positive in about 10% of the population at baseline, is semi-quantitative, and somewhat subjective in pattern determination. The ANA test is helpful for the diagnosis and treatment of some autoimmune

diseases (eg, systemic lupus erythematosus) but is generally not helpful in the absence of a clinically evident autoimmune disorder. As this is a minor point in the paper, I don't know that a change is warranted, but clarification that the comparison is using a different method would be appropriate.

Reviewer #3 (Remarks to the Author):

A major effort in the field has focused on understanding the immunologic mechanisms of spontaneous viral control. Luis M. Molinos-Albert, et al. analyzed the humoral and T cell immune response from post-treatment controllers (PTCs) which are rare HIV infected individuals that underwent ART interruption (ATI) and maintained control of their HIV loads without ART treatment for at least a year. The authors determined that there were two classes of PTCs, stably aviremic PTCs (sPTC) which experienced no transient viremia after ATI and virally exposed PTC (ePTC) which experienced at least one instance of transient viremia. The authors conducted an incredible amount of work dissecting the T cell and humoral immune response between ePTCs, and sPTCs. In addition, for some assays the authors analyzed post-treatment non-controllers (PTNC) both on ART and off ART. The author demonstrated that ePTCs have higher HIV-binding breadth, greater cTh2-like cell number, higher anti-HIV antibody titers, and numerous other phenotypes compared to sPTCs. Additionally, the authors discovered that had ePTC better binding to HIV-infected CEM-NK_r than PTNCs suggesting there may be differences in how the antibodies recognize HIV antigens. The development of neutralization breadth in ePTCs due to transient low viral titer stimulation compared to previous observations in PTNCs which required sustained high viral titer for years, is an interesting observation on the requirements of antigen stimulation and duration in developing neutralization breadth.

While ePTCs having a more robust immune response than sPTCs may be expected due to higher viral loads acting as a boost, the amount of work and characterization of the PTC immune response is commendable. The ePTCs transient low viral stimulation furthers the fields understanding of the requirements for developing neutralization breadth, B cell phenotypes, and T cell phenotypes during HIV infection. This information will be useful in guiding future studies on PTCs and designing future therapeutics or vaccines based on the PTC immune response.

Major issues:

The comparison between antibodies and T cells in sPTC and ePTC is interesting, but expected. sPTCs have controlled viremia and are not experiencing the same high load of antigen that ePTCs are during a viral load spike. This means that ePTCs are more likely to have a robust immune response characterized by more T cells, T_{fh}s, and B cells responses because they are undergoing an active immune response to the antigen in circulation. It is not surprising that the PCA in Figure 2 uses T cell features to separate sPTCs and ePTCs. In Figure 2D, the ePTCs separate with the PTNCs off ART, while the sPTCs separate with the PTNCs on ART suggesting there may be a similarity in their responses due to the circulating HIV virus and antigen. Do the authors believe the higher antibody features in ePTCs compared to sPTCs are due to a better immune response in ePTCs, or due to increased antigen exposure? Comparing sPTC and ePTC responses before ePTCs have had a transient viral spike or after significant time has passed would assist in understanding the relationship.

Do PTNCs lack the cTh2-like cells that were found to correlate with ePTCs? Are cTh2-like cells necessary for the ePTC response, or only a correlate of an ongoing immune response after antigen exposure? If the cTh2-like cells are not necessary for the response, some statements on the roles of

cTh2-like cells in the ePTC response should be emphasized less. Comparing the Tfh response in PTNCs and ePTCs would help elucidate this relationship.

There are several features that are equal or higher in PTNCs compared to ePTCs. This argues that these features are not unique to ePTCs and may not be the cause of sustained viral control. Higher binding to CEM-NKr cells was found to be uniquely upregulated in ePTCs compared to PTNCs, but few other features were. There are several additional phenotyping experiments that would be enhanced by having PTNCs as a control to compare to.

Minor issues:

In the introduction line 52, the authors state “with Fc-dependent effector function potentials” and discussion line 310 “likely promote Fc-mediated”, but never demonstrate that PTC antibodies can activate Fc-dependent functions. Fig 1I. only looks at binding to infected CEM-NKr cells, but does not show any NK cell mediated killing or Fc-dependent function. The authors should be careful with their statements, conduct an antibody-dependent cellular cytotoxicity assay, or perform a similar Fc-dependent assay.

Normally Th1 responses are associated with IgG3 and antiviral responses, and Th2 responses are more associated with IgE responses, which were not characterized. What do the author’s propose that cTh2-like cells are doing to the immune response that assists in controlling HIV?

The authors should include information in the results section about the average time between transient viremia in ePTCs and the analyzed samples. It is unclear of the timing between each event for each sample.

We would like to thank the reviewers for taking the time for a critical reading of our manuscript, and for their thoughtful comments. Please find below our detailed point-by-point response to the three referees.

Reviewer #1 (Remarks to the Author):

Overall, the study includes a comprehensive assessment of multiple immune parameters associated with post-antiretroviral treatment viral control. The study focuses on two important cohorts to address critical questions related to ART-free viral suppression with implications for development of strategies to achieve and maintain viral suppression. Findings indicate immune response differences between post-treatment controllers who have transient, low level viremia [ePTC], relative to sPTC who fully suppress virus. The authors suggest that the ePCT profile in some individuals recalls the functionally-coordinated antibody responses in natural viral controllers, as well as the impact of residual viral replication in promoting higher proportions of HIV-specific memory B cells shown for elite controllers. Not surprisingly, the profile of viral control is complex, since half of the ePTC achieved long-term viral remission in the absence of high-quality humoral immunity.

- *Do the authors have comments about the assays used to profile aspects of immunity, particularly if additional targets/viral protein variants might provide broader overview of the immune profile? Would research studies benefit from more sensitive viral detection methods to relate to results?*

In this study, we performed a comprehensive analysis of the humoral immune response in PTC and PTNC by analyzing up to 45 quantitative and qualitative serological parameters, which now include ADCC activity measurements against HIV-1-infected cells. These analyses focused mostly on HIV-1 Env antibodies, but also included anti-p24 antibody titers in some of the antibody binding assays. These data combined with the cellular immunophenotyping and quantification (memory B-cell and cTfh subsets), and clinico-virological parameters represent, what is to our knowledge, the most comprehensive humoral immunoprofiling study on a cohort of people with HIV-1 under settings of viral control. Additional immune parameters could have been investigated at both, serological and cellular levels (as for instance, the ADCP activity of purified IgGs). We agree that additional measurements may have provided an added value but, we do not feel that other key analyses have been missed. In line with this, it is important to precise that the study was not designed to identify a single correlate of control, but rather for establishing a profile of the humoral response in these individuals: the information obtained already provide a very complete picture allowing the distinction of different profiles among PTC. As mentioned below, other investigators of the Visconti consortium already explore other immunological aspects of the PTC cohort, outside the antibody and B-cell/cTfh components.

The assay used for viral quantification is sensitive. It is not clear whether utilizing the latest most sensitive method would have led to different results. Indeed, based on our previous experience with natural controllers where we have more extensively analyzed that, ultrasensitive tests may help to reveal higher levels of viral replication among individuals who experienced blips (1). However, “non-blippers” did not have detectable viremia even with ultrasensitive techniques (1).

- *Authors point out correctly that the “virus-host interplay among individuals who achieve post-treatment remission is key....to develop novel preventive and therapeutic strategies to combat HIV infection”. What are the implications of the authors’ findings for successful post-ART viral control by therapeutic vaccine strategy; what are implications of the results for development of a protective vaccine regimen?*

We indeed think that uncovering immune correlates of treatment- or naturally-induced viral control provides key information for identifying optimal immune responses to be ideally induced by vaccines. By performing an exhaustive humoral immunoprofiling in PTC, we have provided evidences for the induction of a coordinated-humoral responses of a higher quality in virally exposed PTC, which may help controlling disease progression. We found that episodic low viremia drives neutralizing/cross-neutralizing antibody responses in ePTC as found in highly viremic PTNC, reinforcing the idea that temporary low-antigen exposures may be sufficient to shepherd B-cell maturation and evolution towards antibody neutralization breadth. Importantly, this indicates that sustained high viral loads is not an absolute requirement for developing potent antibody responses, which may be difficult to reproduce by vaccination. Our findings are in agreement with previous observations in natural controllers who developed potent antibody responses. What we observed in HIV-1-infected humans also illustrate the findings in animal models for vaccine design in which the sequential immunization or the slow antigenic delivery of low-dose immunogens induced greater and more functional antibody responses (2, 3). Overall, our results are informative on the role of

transient low antigen stimulation for the development of neutralization breadth, and may therefore be helpful vaccine design. This point is now clarified page 16, line 337-339.

• Is there a relationship between long-term ART-free viral suppression and potential HIV cure? How do the authors assess the study results for cure implications?

This is indeed a very important point. This is addressed by another study looking more specifically at the evolution of the viral reservoir size in PTC, and therefore, could not be included in the present work. It is difficult to determine whether the post-treatment control can be associated with a potential HIV-1 cure. HIV DNA can still be detected in many PTC despite years of control off ART. This probably reflects an optimal equilibrium between the virus and host immune responses, which is maintained for many years. In other cases, the viral reservoirs seem to decrease progressively, until the virus cannot be detected with available techniques and samples. In some of those cases (in occasions with over 20 years of control), also characterized by low activation and inflammation and residual HIV specific responses, we cannot exclude that a status close to HIV cure might have been achieved. However, this is impossible to determine as it is also the case for other documented examples of an extraordinary remission such as after a stem cell transplant. This is why we proposed to define this period of control after treatment interruption as an HIV remission. A manuscript with the description of clinical, immunological and viral evolution of PTC is in preparation

Specific comments.

While important scientifically, the manuscript is challenging to read and omits some essential information. Some examples [not all inclusive] are included below.

• Methods

- Include time frame/years of the original VISCONTI study [n=22] and PRIMO study [n=21]

As requested, the supplementary table 1 now includes the information relative to the follow up of the VISCONTI and PRIMO donors: calendar date of treatment interruption, years from treatment interruption to sampling, and total years from ART initiation.

- Consider harmonizing two supplemental tables into a single table with consistent presentation of data.

Following the reviewer's suggestion, we have combined all clinico-virological information from PTC and PTNC donors in a single table. This set of data is provided in the supplementary table 1.

- Use group designations consistently. Table S1. PTC/sPTC, post-treatment controllers [n=10]; ePTC/rebounders, in text, PTC rebounders [n=3]; ePTC, virally exposed PTC [n=9]; Table S2. PTNC, post-treatment non-controllers [n=21]. Note that the eART acronym in the manuscript uses "e" for "early"

We apologize for the confusion. Group designations in the text, figures and tables have been corrected for consistency. For clarity, ePTC who resumed to ART by the time or before the analysis (070001, 200001 and 216001) are now identified by a footnote in the supplementary table 1, and the abbreviation eART, initially called only once in the text, has been deleted (page 3, lines 36-38).

- What types of ART were received? While responses are classified by CD4 and VL, the type of treatment/ART needs to be included as a variable.

This is indeed an important information. As requested, ART regimens received by PTC and PTNC donors are now specified in the supplementary table 1.

- What is definition of early treatment in VISCONTI cohort [Table S1]? While early treatment is 26 to 100 days post-infection in PRIMO cohort [Table S2], the Early Treatment designation of + or - in Table S1 is inadequate.

We apologize for the confusion regarding the discordant definition of early treatment between both cohorts. With the exception of two donors (005002 and 038001, page 5 lines 58-60), all PTC initiated effective ART during primary HIV-1 infection. As requested, we have now included the estimated time for ART initiation (in days post-infection) for PTC as it was initially described for PTNC. Please see supplementary table 1.

- Statistical analysis methods have no set point for significant P-value. Did statistical methods include corrections for multiple comparisons? What statistical software were used?

Considering the explorative nature of our study, all statistical analyses were performed with a descriptive purpose. Thus, the levels of significance of the multiparametric associations were initially not formally adjusted for multiple comparisons. We believe that while correcting for multiple comparisons reduces the chances of making type I errors (findings of false significance), it also increases the chances of type II errors (false negatives), the latter being more problematic than the former in descriptive studies as it would conceal some findings worth to be explored by future studies. Nevertheless, to address the reviewer's concern and to provide an assessment for multiple testing to the readers, Bonferroni-corrected p -value thresholds have been implemented in the figure panels 3A, 4A, 5D, 5G, 5L, 5O, 7A, 7B and S6A. As detailed in figure legends of the revised manuscript, p -values below the $p_{\text{Bonferroni}}$ are highlighted. In addition, to provide the full set of data for an appropriate statistical assessment, detailed correlation results of the multiparametric analyses (numeric p -values, Spearman rho and 95% confidence intervals) have been included in a new file designated as Supplementary data 1. The Methods section describing the "Statistical analyses" has been implemented accordingly.

- Any evidence for superinfection within either of the cohorts? Were viral genotypes in PTNC or ePTC evaluated?

We have not found any evidences of superinfection within the historical medical files of the donors. Regarding the viral genotyping of the donors, we do not have this information for all the donors yet, but again the preliminary sequencing data and phylogenetic analyses do not argue for superinfection events. Investigators in the ANRS VISCONTI Study group are working on this matter, among other clinically relevant issues, in a manuscript currently under preparation; we therefore consider this analysis is beyond the scope of the present study.

- Cell-associated virus quantitation was not included in the assessment. While the assay may be unfeasible for the samples from the cohorts included in the reported study, do the authors consider such assessments of value for future studies?

This is clearly a very relevant point for the ANRS VISCONTI consortium given that the evolution of the viral reservoir in PTC could indeed provide important clues for the research towards HIV-1 remission. As explained above, we consider that these analyses, being assessed by our collaborators and part of an independent manuscript under preparation, are out of the scope of the present study.

• Edits

- The second section of Results extends three pages in the manuscript [p.5, line 79 thru p.8, line 158] and includes data from two figures and three supplemental figures, each with multiple data panels. The narrative would benefit from subheadings.

We agree with the reviewer's comment. Accordingly, and to also comply with the *Nature Communications* editorial policies, this section has been sub-divided into two parts for enhancing clarity to the readers: "Comprehensive serological antibody profiling of PTC and PTNC" (lines 79-115) followed by "ePTC develop robust functional antibody responses" (lines 117-172).

- Data included in Figures should be reviewed to reformat or move to supplemental. Some examples: bars in Fig 1G are essentially illegible for red/blue distribution.

We apologize for the lack of clarity. The panel 1G has been modified to improve sharpness and comprehensibility.

; Fig 1 includes variant designations in different panels for the same individual [one example: Fig 1F, 84001, but 084001 in Fig 1H and Fig 1I]

We apologize for the typos; we have rectified the donor codes for homogeneity / harmonization across tables, figures and in the text of the revised manuscript.

; blue color scale is difficult to assess in many data panels;

We have increased the size of the bar scales accompanying the multiparameter matrixes to facilitate the assessment.

Fig S6C: Is the method/citation for t-SNE-derived statistical heatmaps included in the Methods?

To visualize B-cell subpopulations analyzed by flow cytometry, we used the Barnes-Hut t-distributed Stochastic Neighbor Embedding (t-SNE) approximation, by applying a perplexity value of 200 and 1000 iterations. This was initially described at the end of the *Flow cytometric immunophenotyping* description in

the *Methods* section. We have transferred it into the *Statistics* section of the revised manuscript, and added the corresponding reference describing the algorithm.

Fig S1E, what is the value of the panel?

In Fig S1E, IgA antibodies purified from PTC's sera showed no neutralization activity. The panel indicates no neutralization detected. A legend has been included for clarification.

- Review English language use.

The manuscript was edited by a native English speaker, and the grammar syntax errors have been corrected.

Reviewer #2 (Remarks to the Author):

This is a manuscript describing an in-depth characterization of a cohort of HIV-1 infected persons who underwent a structured treatment interruption and who were then found to control their viremia. The cohort is broken into those who had strict control and those who had blips, with the finding that those who had blips developed antibodies with breadth against various HIV-1 strains. This is a very dense paper with many multi-panel figures indicating the large amount of work that was done. Overall, the manuscript does a good job of showing that there are differences between those who have blips and those who do not.

The key take home of the paper—that people who have blips develop different responses than those who are stably aviremic—is important and contributes to our understanding of these responses, and as noted could help guide future vaccine design. I have one concern and one comment.

We thank the reviewer for her/his overall positive feedback on our study.

Concern

There is a massive amount of data in the paper, and many, many statistical tests are performed to compare groups and conditions. While the tests appear to be appropriate and properly applied, there is no discussion of correction for multiple comparisons and how this affects the analysis. I do not suggest that additional experiments be performed, but this confounder—ie, the fact that by performing so many tests a number of “statistically significant” results at $p < 0.05$ would be expected—should be discussed. Ideally, the analysis would include correction and discussion of corrected and uncorrected p -values presented.

We agree with the reviewer's comment. As mentioned to Reviewer's 1, we have now included the detailed results for the multiparametric analyses including multiple comparisons in the “Supplementary data 1” file, and implemented the figures with the Bonferroni-corrected p -value thresholds.

Comment

A minor comment is made that the cohort was tested for autoantibodies by ANA and were found to be negative, contrasting these findings with those of a prior study of HIV-1-infected persons where increased autoantibody frequency was found among those who developed broad neutralizing antibodies. The paper cited did not use ANA as a measure of autoantibodies but rather used the AtheNA MultiLyte test which provides quantitation of autoantibody binding to a range of autoantigens. In full disclosure, I was a coauthor on the cited study, and we specifically did not use the ANA test because of the challenges of interpreting that test in this setting. The ANA test is positive in about 10% of the population at baseline, is semi-quantitative, and somewhat subjective in pattern determination. The ANA test is helpful for the diagnosis and treatment of some autoimmune diseases (eg, systemic lupus erythematosus) but is generally not helpful in the absence of a clinically evident autoimmune disorder. As this is a minor point in the paper, I don't know that a change is warranted, but clarification that the comparison is using a different method would be appropriate.

We thank the reviewer for bringing up this issue. We do agree that HEp-2 IFA is not a very sensitive and quantitative assay to detect autoantibody levels, particularly in non-autoimmune individuals. We also agree that accurately comparing data across studies necessitates performing the testing of samples with the same assays. Therefore, we had initially planned using a Bioplex system (MILLIPLEX MAP Human Autoimmune Autoantibody Panel) to evaluate the cross-reactivity of the PTC and PTNC IgGs to self-antigens. However, the provider informed us that the kit was back-ordered (most likely to COVID-19 supplying issues) with an

estimated delivery date beginning of 2022, which is incompatible with a timely re-submission. Thus, as an alternative, we have evaluated the IgG autoreactivity by ELISA using a panel of 10 self-antigens including Jo-1, PR-3, Histone H2B, Hsp70, collagen I, cardiolipin, C1q, dsDNA, insulin and thyroglobulin. The results are now presented in Fig. S2 (panel C to G). These data showed that the levels of cross-reactive IgG antibodies to self-antigens do not statistically differ between sPTC and ePTC, but were significantly higher in PTNC than in ePTC (Fig.S2E). Of note, viral loads in PTNC had no effect on the magnitude of self-reactivity (Fig.S2G). Strikingly, however, PTC with neutralization score above 10 displayed more elevated IgG cross-reactivity than those below 10 (Fig.S2F). We now describe in the revised manuscript the cross-reactivity to self-antigens in PTC in light of these new results (page 7, lines 126-133).

Reviewer #3 (Remarks to the Author):

A major effort in the field has focused on understanding the immunologic mechanisms of spontaneous viral control. Luis M. Molinos-Albert, et al. analyzed the humoral and T cell immune response from post-treatment controllers (PTCs) which are rare HIV infected individuals that underwent ART interruption (ATI) and maintained control of their HIV loads without ART treatment for at least a year. The authors determined that there were two classes of PTCs, stably aviremic PTCs (sPTC) which experienced no transient viremia after ATI and virally exposed PTC (ePTC) which experienced at least one instance of transient viremia. The authors conducted an incredible amount of work dissecting the T cell and humoral immune response between ePTCs, and sPTCs. In addition, for some assays the authors analyzed post-treatment non-controllers (PTNC) both on ART and off ART. The author demonstrated that ePTCs have higher HIV-binding breadth, greater cTh2-like cell number, higher anti-HIV antibody titers, and numerous other phenotypes compared to sPTCs. Additionally, the authors discovered that had ePTC better binding to HIV-infected CEM-NK_r than PTNCs suggesting there may be differences in how the antibodies recognize HIV antigens. The development of neutralization breadth in ePTCs due to transient low viral titer stimulation compared to previous observations in PTNCs which required sustained high viral titer for years, is an interesting observation on the requirements of antigen stimulation and duration in developing neutralization breadth.

While ePTCs having a more robust immune response than sPTCs may be expected due to higher viral loads acting as a boost, the amount of work and characterization of the PTC immune response is commendable. The ePTCs transient low viral stimulation furthers the fields understanding of the requirements for developing neutralization breadth, B cell phenotypes, and T cell phenotypes during HIV infection. This information will be useful in guiding future studies on PTCs and designing future therapeutics or vaccines based on the PTC immune response.

We are thankful to the referee for her/his positive general comments.

Major issues:

The comparison between antibodies and T cells in sPTC and ePTC is interesting, but expected. sPTCs have controlled viremia and are not experiencing the same high load of antigen that ePTCs are during a viral load spike. This means that ePTCs are more likely to have a robust immune response characterized by more T cells, T_Hs, and B cells responses because they are undergoing an active immune response to the antigen in circulation. It is not surprising that the PCA in Figure 2 uses T cell features to separate sPTCs and ePTCs. In Figure 2D, the ePTCs separate with the PTNCs off ART, while the sPTCs separate with the PTNCs on ART suggesting there may be a similarity in their responses due to the circulating HIV virus and antigen. Do the authors believe the higher antibody features in ePTCs compared to sPTCs are due to a better immune response in ePTCs, or due to increased antigen exposure? Comparing sPTC and ePTC responses before ePTCs have had a transient viral spike or after significant time has passed would assist in understanding the relationship.

We thank the reviewer for raising this important point. We agree that the “silent immunoprofiles” common to sPTC and PTNC under ART is to be linked with an absence of ongoing viral replication. Conversely, that the development of a higher-in-magnitude humoral response to HIV-1 in ePTC and PTNC off ART is due to the antigenic stimulation. However, it is important to highlight that ePTC did not experience high and prolonged antigen exposures, but only transient episodes of low viremia. Thus, finding similar

immunoprofiles between ePTC under post-treatment viral control and PTNC post-TI who experienced very high and durable viremia was not expected. Importantly, one third of the ePTC showed cross-seroneutralization including donor 005002 who developed an elite neutralizer-type of profile. As discussed in the manuscript, we propose that these temporary low antigen exposures are sufficient to support B-cell maturation and evolution towards cross-neutralizing antibodies in ePTC. This is likely facilitated by a better early ART-dependent preservation of the immune compartments including cTfh cells. It is thus tempting to speculate that equivalent outcomes in regard to the humoral response to HIV-1 Env would be expected in sPTC upon similar antigenic stimulations, at least in a fraction of them. As indicated above, we have included the time-frame between the last transient viremic episode >50 copies/ml and sampling dates for ePTC in the supplementary table 1. This shows, considering the time at which the analyses were made, that the comparison between ePTC and sPTC subgroups is appropriate. Unfortunately, due to the lack of appropriate samples, we could not make the suggested comparison between sPTC, and ePTC before and after viral blips in all donors. We did do so for the available follow-up samples (under ART and post-ATI), which includes only 4 ePTC and 1 sPTC (Fig. 1F). Although it is very difficult to draw conclusions on these analyses regarding whether ePTC had higher/better antibody B-cell response to start with, which might not be the case, we clearly observed, as mentioned in the manuscript, that viral exposures drive the enhanced coordinated humoral response to HIV-1 Env in ePTC.

Do PTNCs lack the cTh2-like cells that were found to correlate with ePTCs? Are cTh2-like cells necessary for the ePTC response, or only a correlate of an ongoing immune response after antigen exposure? If the cTh2-like cells are not necessary for the response, some statements on the roles of cTh2-like cells in the ePTC response should be emphasized less. Comparing the Tfh response in PTNCs and ePTCs would help elucidate this relationship.

We agree with the referee's concern in establishing direct links between the most prevalent cTfh cell subsets and antibody seroprofiles identified in PTC. While the B-cell help capacity of cTfh has been extensively evaluated *in vitro* (4, 5), previous studies demonstrated a clonal relationship between tonsillar GC Tfh and blood cTfh co-expressing PD1 and ICOS, which were activated and expanded upon antigen recall (6). Therefore, activated cTfh cells represent important surrogate markers of germinal center activity. In this study, we unveiled certain associations between viremic exposures, specific cTfh and memory B-cell subsets, and potent serological responses. We found that CXCR5⁺ICOS⁺, CXCR5⁺PD1^{hi} and CXCR5⁺PD1^{hi}ICOS⁺ cTfh cells were mobilized in ePTC, associated with a shift in blood memory B-cell subset distribution and potent antibody responses, reminiscent of previous studies supporting blood cTfh as surrogate markers of ongoing GC reactions. As the reviewer points correctly, humoral responses in ePTC were dominated by higher frequencies of CXCR3⁺CCR6⁻ Th2-like cTfh cells. Moreover, we showed correlation patterns with ICOS⁺, PD1^{hi} and PD1^{hi}ICOS⁺ both within CXCR3⁺ and CXCR3⁻ cTfh subsets, and we also found a preferential expansion of PD1^{hi}CXCR3⁺ cTfh associated with higher recurrence of transient viremia, and certain B-cell and serological parameters. Therefore, not only Th2-like cTfh were preferentially associated with ePTC, and distinct cTfh subsets may indeed contribute to higher-in-magnitude polyfunctional humoral profiles. We also concluded that the cTfh compartment in PTC is not dysregulated, but activated upon recalls by low antigenemia. Our findings build on published data describing diverse cTfh subsets (Th1- and Th2-like) involved in the induction of functional antibodies during HIV-1 infection. All these associations were well-described in the manuscript.

We fully agree with the reviewer that comparing cTfh sub-populations in PTNC and ePTC would have been helpful to potentially elucidate the relationships between cTfh and antibody B-cell response. Unfortunately, there were no PBMCs bio-banked for these donors/timepoints in the ANRS Primo cohort, which precludes analyzing the cTfh and memory B-cell phenotypes in the PTNC donors. Nonetheless, considering that Th2-like cTfh (PD1^{hi}CXCR3⁺CXCR5⁺) have been shown to be expanded in individuals developing HIV-1 neutralizing breadth (5), a similar profile in PTNC neutralizers as in ePTC would be expected.

There are several features that are equal or higher in PTNCs compared to ePTCs. This argues that these features are not unique to ePTCs and may not be the cause of sustained viral control. Higher binding to CEM-NKr cells was found to be uniquely upregulated in ePTCs compared to PTNCs, but few other features were. There are several additional phenotyping experiments that would be enhanced by having PTNCs as a control to compare to.

One of the key findings of this study is that by only experiencing few transient episodes of low viremia, a substantial fraction of ePTC develop potent antibody responses including neutralizing breadth that, as the reviewer points, they were similar or higher to those found in PTNC, who experienced very high levels of viremia after treatment interruption. These findings, congruent with those made on Elite controllers (7–9), are highly relevant for HIV control/remission and have direct implications in vaccine development as showing that developing potent immune responses occur with transient and low viremia exposures under viral control. We agree with the reviewer that immunophenotyping/quantifying memory B cells and cTfh subpopulations in PTNC, and compare the data to those of ePTC would have provided valuable information. Unfortunately, as mentioned above, we could not get access to the PBMCs from these donors as only serum samples were available, precluding *per se* any additional experiments. Blood cells are only collected in a sub-study of the ANRS PRIMO cohort.

Minor issues:

In the introduction line 52, the authors state “with Fc-dependent effector function potentials” and discussion line 310 “likely promote Fc-mediated”, but never demonstrate that PTC antibodies can activate Fc-dependent functions. Fig 11. only looks at binding to infected CEM-NKr cells, but does not show any NK cell mediated killing or Fc-dependent function. The authors should be careful with their statements, conduct an antibody-dependent cellular cytotoxicity assay, or perform a similar Fc-dependent assay.

We agree with the reviewer that although the binding to infected cells represents a good surrogate for Fc-dependent effector functions of Env antibodies such as NK-mediated ADCC activity (10, 11), it does not constitute a demonstration *per se*. To address this point, we performed new experiments to determine the ADCC activity of purified IgGs from PTC and PTNC through the primary human NK cell-mediated elimination of target cells infected either with YU2 or CH058 virus. Our data presented in the new Figure 2 (also including the original neutralization and infected-cell binding data), as well as in a new supplementary Figure 4. These novel results described in the revised manuscript (page 8, lines 145-151) show, as expected, that the ADCC activity of serum IgGs, measured as the percent of eliminated infected cells, is statistically higher in ePTC compared to sPTC (Fig. 2D). We found significant correlations between the levels of ADCC and binding to cells infected for YU-2 and CH058 viruses in PTC, and only for YU-2 in PTNC (Fig. 2E). Interestingly, %ADCC against CH058-infected targets correlated with the global neutralization score in PTC and PTNC (Fig. 2F). Similarly, a correlation between %ADCC against YU2-infected targets and the neutralization score was also revealed but only for PTC (Fig. 2F). These results are discussed in the text (page 15, lines 322-326).

Normally Th1 responses are associated with IgG3 and antiviral responses, and Th2 responses are more associated with IgE responses, which were not characterized. What do the author’s propose that cTh2-like cells are doing to the immune response that assists in controlling HIV?

According to the current knowledge on human cTfh cells, the distinction between Th1- and Th2-like subpopulations can be established based on the differential surface expression of CXCR3 and CCR6 chemokine receptors. While Th1-like cTfh are CCR6⁻CXCR3⁺ expressing T-bet transcription factor and Th1 cytokine interferon- γ , Th2-like cTfh cells are CCR6⁺CXCR3⁻ expressing GATA-3 and Th2-like cytokines IL-4, IL-5 and IL-13. As the reviewer correctly points out, Th2-like cTfh have been associated with IgE responses, but also with IgG secretion (4, 12). Indeed, numerous studies have shown that Th2-like cTfh can efficiently activate naive B-cells and induce IgG-class switching (4, 5, 13, 14). In contrast, Th1-like cTfh, while involved in B-cell stimulation in settings of low antigenemia or poor vaccine efficacy, appear unable to promote primary antibody responses and isotype class-switching, and are thus often referred as poor helpers (15, 16). As discussed above, and in the manuscript, the direct implication of cTfh in supporting PTC viral control would be directly linked to their role in promoting B-cell activation and proliferation to produce long lasting neutralizing antibody responses. As mentioned in the main text, the fact that recurrent low viral exposures did not lead to viral rebound in ePTC developing neutralization breadth could indicate a role of antibodies in the viral control. Therefore, we propose that cTfh, and particularly Th2-like cTfh, represent an essential immune component in developing polyfunctional and neutralizing HIV-1 antibody responses in PTC.

The authors should include information in the results section about the average time between transient viremia in ePTCs and the analyzed samples. It is unclear of the timing between each event for each sample.

Thank you for the suggestion. We now specify in the results section (page 5, lines 68-70), and in the supplementary table 1 of the revised manuscript, the average time between the last viral load (>50 copies/ml) and sampling dates.

References

1. E. Canouï, C. Lécroux, V. Avettand-Fenoël, M. Gousset, C. Rouzioux, A. Saez-Cirion, L. Meyer, F. Boufassa, O. Lambotte, N. Noël, A subset of extreme human immunodeficiency virus (HIV) controllers is characterized by a small HIV blood reservoir and a weak T-cell activation level. *Open Forum Infect. Dis.* **4**, 1–8 (2017).
2. A. Escolano, J. M. Steichen, P. Dosenovic, D. R. Burton, W. R. Schief, M. C. Nussenzweig, A. Escolano, J. M. Steichen, P. Dosenovic, D. W. Kulp, J. Golijanin, D. Sok, Sequential Immunization Elicits Broadly Neutralizing Anti-HIV-1 Antibodies in Ig Knockin Mice Article Sequential Immunization Elicits Broadly Neutralizing Anti-HIV-1 Antibodies in Ig Knockin Mice. *Cell*. **166**, 1445-1458.e12 (2016).
3. K. M. Cirelli, D. G. Carnathan, B. Nogal, J. T. Martin, O. L. Rodriguez, A. A. Upadhyay, C. A. Enemuo, E. H. Gebru, Y. Choe, F. Viviano, C. Nakao, M. G. Pauthner, S. Reiss, C. A. Cottrell, M. L. Smith, R. Bastidas, W. Gibson, A. N. Wolabaugh, M. B. Melo, B. Cossette, V. Kumar, N. B. Patel, T. Tokatlian, S. Menis, D. W. Kulp, D. R. Burton, B. Murrell, W. R. Schief, S. E. Bosinger, A. B. Ward, C. T. Watson, G. Silvestri, D. J. Irvine, S. Crotty, Slow Delivery Immunization Enhances HIV Neutralizing Antibody and Germinal Center Responses via Modulation of Immunodominance. *Cell*. **177**, 1153-1171.e28 (2019).
4. R. Morita, N. Schmitt, S. E. Bentebibel, R. Ranganathan, L. Bourdery, G. Zurawski, E. Foucat, M. Dullaers, S. K. Oh, N. Sabzghabaei, E. M. Lavecchio, M. Punaro, V. Pascual, J. Banchereau, H. Ueno, Human Blood CXCR5+CD4+T Cells Are Counterparts of T Follicular Cells and Contain Specific Subsets that Differentially Support Antibody Secretion. *Immunity*. **34**, 108–121 (2011).
5. M. Locci, C. Havenar-Daughton, E. Landais, J. Wu, M. a. Kroenke, C. L. Arlehamn, L. F. Su, R. Cubas, M. M. Davis, A. Sette, E. K. Haddad, P. Poignard, S. Crotty, Human circulating PD-1+CXCR3-CXCR5+ memory Tfh cells are highly functional and correlate with broadly neutralizing HIV antibody responses. *Immunity*. **39**, 758–769 (2013).
6. A. Heit, F. Schmitz, S. Gerdt, B. Flach, M. S. Moore, J. A. Perkins, H. S. Robins, A. Aderem, P. Spearman, G. D. Tomaras, S. C. De Rosa, M. J. McElrath, Vaccination establishes clonal relatives of germinal center T cells in the blood of humans. *J. Exp. Med.* **214**, 2139–2152 (2017).
7. J. F. Scheid, H. Mouquet, N. Feldhahn, M. S. Seaman, K. Velinzon, J. Pietzsch, R. G. Ott, R. M. Anthony, H. Zebroski, A. Hurley, A. Phogat, B. Chakrabarti, Y. Li, M. Connors, F. Pereyra, B. D. Walker, H. Wardemann, D. Ho, R. T. Wyatt, J. R. Mascola, J. V. Ravetch, M. C. Nussenzweig, Broad diversity of neutralizing antibodies isolated from memory B cells in HIV-infected individuals. *Nature*. **458**, 636–40 (2009).
8. N. T. Freund, H. Wang, L. Scharf, L. Nogueira, J. a Horwitz, Y. Bar-On, J. Golijanin, S. a Sievers, D. Sok, H. Cai, J. C. C. Lorenzi, A. Halper-Stromberg, I. Toth, A. Piechocka-Trocha, H. B. Gristick, M. J. Van Gils, R. W. Sanders, L.-X. Wang, M. S. Seaman, D. R. Burton, A. Gazumyan, B. D. Walker, A. P. West, P. J. Bjorkman, M. C. Nussenzweig, H I V Coexistence of potent HIV-1 broadly neutralizing antibodies and antibody-sensitive viruses in a viremic controller. *Sci. Transl. Med.* **2**, 1–14 (2017).
9. J. F. Scheid, H. Mouquet, B. Ueberheide, R. Diskin, F. Klein, T. Y. K. Oliveira, J. Pietzsch, D. Fenyo, A. Abadir, K. Velinzon, A. Hurley, S. Myung, F. Boulad, P. Poignard, D. R. Burton, F. Pereyra, D. D. Ho, B. D. Walker, M. S. Seaman, P. J. Bjorkman, B. T. Chait, M. C. Nussenzweig, Sequence and Structural Convergence of Broad and Potent HIV Antibodies That Mimic CD4 Binding. *Science*. **333**, 1633–1637 (2011).
10. T. Bruel, F. Guivel-Benhassine, S. Amraoui, M. Malbec, L. Richard, K. Bourdic, D. A. Donahue, V. Lorin, N. Casartelli, N. Noël, O. Lambotte, H. Mouquet, O. Schwartz, Elimination of HIV-1-infected cells by broadly neutralizing antibodies. *Nat. Commun.* **7**, 10844 (2016).
11. T. Bruel, F. Guivel-Benhassine, V. Lorin, H. Lortat-Jacob, F. Baleux, Lack of ADCC Breadth of Human Nonneutralizing Anti-HIV-1 Antibodies. *J. Virol.* **91**, 1–19 (2017).
12. H. Ueno, J. Banchereau, C. G. Vinuesa, Pathophysiology of T follicular helper cells in humans and mice. *Nat. Immunol.* **16**, 142–152 (2015).
13. K. L. Boswell, R. Paris, E. Boritz, D. Ambrozak, T. Yamamoto, S. Darko, K. Wloka, A. Wheatley, S. Narpala, A. McDermott, M. Roederer, R. Haubrich, M. Connors, J. Ake, D. C. Douek, J. Kim, C. Petrovas, R. A. Koup, Loss of Circulating CD4 T Cells with B Cell Helper Function during Chronic HIV Infection. *PLoS Pathog.* **10**, 1–14 (2014).
14. R. Cubas, J. van Grevenynghe, S. Wills, L. Kardava, B. H. Santich, C. M. Buckner, R. Muir, V. Tardif, C. Nichols, F. Procopio, Z. He, T. Metcalf, K. Ghneim, M. Locci, P. Ancuta, J.-P. Routy, L. Trautmann, Y. Li, A. B. McDermott, R. A. Koup, C. Petrovas, S. A. Migueles, M. Connors, G. D. Tomaras, S. Moir, S. Crotty, E. K. Haddad, Reversible Reprogramming of Circulating Memory T Follicular Helper Cell Function during Chronic HIV Infection. *J. Immunol.* **195**, 5625–5636 (2015).
15. N. Schmitt, S. E. Bentebibel, H. Ueno, Phenotype and functions of memory Tfh cells in human blood. *Trends Immunol.* **35**, 436–442 (2014).
16. S. Crotty, T Follicular Helper Cell Biology: A Decade of Discovery and Diseases. *Immunity*. **50**, 1132–1148 (2019).

REVIEWER COMMENTS

Reviewer #1 (Remarks to the Author):

I reviewed the revised materials and commend the authors for their comprehensive and thoughtful responses and revisions. I have no additional comments or questions. The revised manuscript is significantly improved and includes results that are significant for the field.

Reviewer #2 (Remarks to the Author):

The authors appear to have addressed the issues raised in prior review. I have no additional concerns.

REVIEWERS' COMMENTS

Reviewer #1 (Remarks to the Author):

I reviewed the revised materials and commend the authors for their comprehensive and thoughtful responses and revisions. I have no additional comments or questions. The revised manuscript is significantly improved and includes results that are significant for the field.

Reviewer #2 (Remarks to the Author):

The authors appear to have addressed the issues raised in prior review. I have no additional concerns.

We thank the reviewers for their positive feedback. We wish to thank them again for their insightful comments, which helped us improving the manuscript.